# Evaluation of Virtual Stain Multiplexed CD68 for Macrophage Detection in NSCLC PD-L1 Slides

**Elad Arbel**                                         ELAD.ARBEL@AGILENT.COM
**Oded Ben-David** (iD)                                ODED.BEN-DAVID@AGILENT.COM
**Itay Remer**
**Amir Ben-Dor**
**Daniela Rabkin**
*Agilent Research Laboratories, Tel Aviv, Israel*

**Sarit Aviel-Ronen**
*Adelson School of Medicine, Ariel University, Ariel, Israel*

**Frederik Aidt**
**Tine Hagedorn-Olsen**
**Lars Jacobsen**
*Agilent Technologies, Glostrup, Denmark*

**Kristopher Kersch**
**Jim Christian**
**Quyen Nguyen**
*Agilent Technologies, Carpinteria, US*

**Anya Tsalenko**
*Agilent Research Laboratories, Santa Clara, US*

**Editors:** Accepted for publication at MIDL 2025

## Abstract

Manual reading of tissue slides by pathologists serves both as a foundation for clinical decision-making and as a source of ground truth for training artificial intelligence (AI) models. However, challenges such as inter-observer variability, limited tissue availability, and complex annotation tasks often compromise reliability and scalability. This study exemplifies a broader trend in pathology: leveraging virtual staining and other AI-based methodologies to address these challenges. We applied virtual stain multiplexing to a challenging annotation task - macrophage identification in non-small cell lung cancer tissue PD-L1 IHC stains, demonstrating its ability to improve pathologist performance and inter-observer agreement. In six challenging regions selected from 49 curated whole slide images, virtual staining significantly increased macrophage detection consistency, with Fleiss' kappa improving from -0.1 to 0.62, and enhanced overall accuracy, with the F1 score increasing from 0.13 to 0.65. These results highlight the potential use of AI-based virtual staining to assist pathologists reading slides, thereby improving consistency, enhancing accuracy, and alleviating the dependence on additional costly staining. Virtual stain multiplexing demonstrates a generalizable approach to improving pathologist performance through measurement-based AI tools, addressing broader needs for reproducibility and efficiency in diagnostic pathology.

**Keywords:** Immunohistochemistry, Virtual Stain, Multiplexing, Deep Learning, Macrophages, NSCLC, PD-L1, AI, Agreement, Pathology

## 1. Introduction

A central challenge in clinical pathology is the need to extract ever-increasing information from progressively smaller tissue samples (Hirsch et al., 2010). This challenge is amplified by the growing demand for personalized medicine (Hirsch et al., 2010), which often necessitates application of a broad range of immunohistochemical (IHC) stains onto multiple consecutive tissue sections from each patient sample. This process is time-consuming and constrained by limited tissue availability, especially in the case of small biopsies, and incurs significant costs in terms of labor and reagents. Furthermore, consecutive tissue sections necessarily contain different cells, adding uncertainty to IHC status or cell type determinations obtained across adjacent sections. While research methodologies, such as fluorescent IHC multiplexing (Hoyt, 2021) or mass spectrometry imaging (Keren et al., 2019), offer potential solutions, they are not yet practical for routine diagnostic use.

Digital pathology provides alternative solutions to this challenge, such as virtual staining (VS) (Imran et al., 2024; Latonen et al., 2024), a novel approach leveraging generative adversarial networks (GANs) to infer staining patterns directly from existing hematoxylin and eosin (H&E)-stained tissue without the need for additional physical staining (Owkin, 2020; de Haan et al., 2021; Pati et al., 2024). This method has evolved to allow inference from unstained tissue (Rivenson et al., 2019; Zhang et al., 2020; Pradhan et al., 2021), including virtual HER2 IHC (Bai et al., 2022; Kapil et al., 2021).

Another critical challenge in pathology is low agreement between pathologists on some clinically relevant tasks (Robert et al., 2023; Han et al., 2022). Even for relatively straightforward tasks, such as scoring HER2 IHC 3+ specimens, inter-pathologist concordance is often suboptimal, potentially leading to inconsistent patient treatment decisions (Robbins et al., 2023). Addressing this issue, new quantitative measurement-based methods have been proposed to replace traditional subjective slide interpretations (Li et al., 2024; Aidt et al., 2025). It has been suggested that the deterministic nature of computational image-processing and AI-based methods can help alleviate subjectivity and improve reproducibility in these contexts (Baxi et al., 2022; Iyer et al., 2024; Pulaski et al., 2024).

However, the development of AI models requires reliable, large-scale ground truth data, which can be costly when derived from manual pathologist annotations. Low agreement among annotators often necessitates employing multiple pathologists per sample and using majority vote or consensus as ground truth, further compounding costs. A recent study comparing AI models for HER2 scoring showed that agreement among automated models was similar to agreement among pathologists(McKelvey et al., 2024), potentially due to the models being trained on subjective, manually scored data. Using measurement-based ground truth for training AI models could overcome these limitations and lead to more reliable diagnostic tools.

VS methods trained with measurement-based ground truth offer a potential solution, providing reliable information to improve pathologist accuracy and reproducibility. For instance, in PD-L1 IHC 22C3 pharmDx stained non-small cell lung cancer (NSCLC) tissues, macrophage detection presents a notable challenge, due to their morphological similarity to tumor cells (Paces et al., 2022; Tsao et al., 2018). Beck et al. (Beck et al., 2019) reported similarly low agreement for macrophage detection in urothelial carcinoma, with inter-observer correlation as low as 0.287. The PD-L1 IHC 22C3 pharmDx assay is FDA-

approved for detecting PD-L1 protein expression in NSCLC tissue, scored using the tumor proportion score (TPS), which excludes immune cells such as macrophages from the calculation (Agilent Technologies, 2021). In near threshold cases, misidentification of tumor cells and macrophages can lead to change in treatment decision. Differentiation of tumor cells and macrophages could be improved by multiplex IHC staining, such as combined staining of PD-L1 with a tumor marker (Mercier et al., 2023), or with a high signal-to-noise macrophage marker like CD68 (Frafjord et al., 2020), but this is not routinely available in clinical settings.

To address these challenges, we recently introduced a method for training AI-based virtual stain multiplexing (VSM) models using same-section sequential IHC staining as a measurement-based ground truth (Ben-David et al., 2024). In this study, we present an in-depth evaluation of this approach, demonstrating its efficacy in creating VSM CD68 to assist pathologists in detecting macrophages in tumor microenvironment of PD-L1 IHC 22C3 pharmDx-stained NSCLC samples. Our findings show that VSM significantly improves both pathologist performance and inter-pathologist agreement on this challenging task.

## 2. Methods

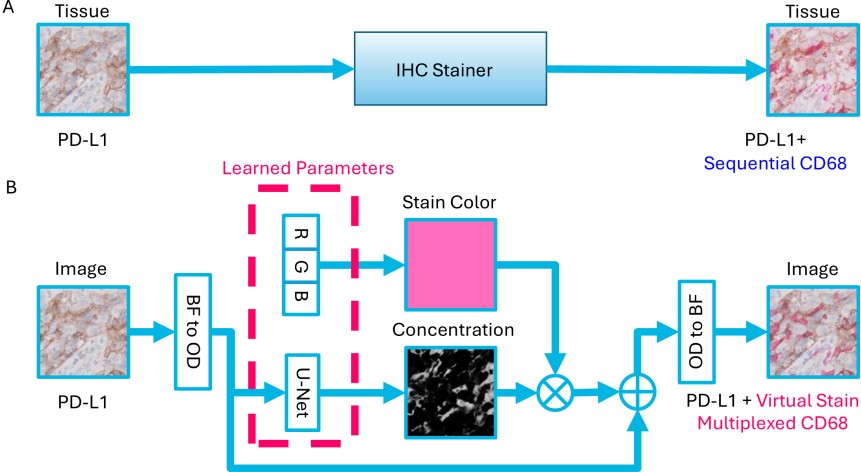

Figure 1: Workflow for Sequential slide staining and Virtual Staining processes. (A) A PD-L1 IHC stained slide sequentially stained with CD68 IHC on the same section to generate a PD-L1 and PDL1+CD68 image pair. (B) A virtually stained multiplexed PD-L1 + CD68 image is generated from a PD-L1 IHC stained WSI.

Our VSM approach (Ben-David et al., 2024) involves training a neural network to simulate sequential stain multiplexing, enabling virtual multiplexing without physical restaining, as illustrated in Figure 1. The technique models the physical Beer-Lambert law, which governs the optical absorption of stained tissue in scanned histopathological images.

In the sequential multiplexing workflow, illustrated in Figure 1A, a PD-L1 immunohistochemistry (IHC) stained tissue undergoes a sequential staining with an additional CD68 IHC marker applied to the same tissue section. The VSM workflow is illustrated in Figure 1B. A U-Net convolutional neural network (CNN) forms the backbone of our virtual stain generation process. The input to the U-Net is a patch extracted from the PD-L1 stained brightfield (BF) RGB image, converted to optical density (OD) space to align with the Beer-Lambert law. The network predicts a concentration map of the CD68 stain, which is then multiplied by a learned RGB color vector, generating a virtual CD68 stain patch. To produce the final VSM image, the virtual CD68 patch is combined with the input PD-L1 patch in OD space and converted back to RGB space, resulting in a synthetically stained PD-L1 + virtual CD68 image pair.

The dataset used for training and evaluation is detailed in (Ben-David et al., 2024). The study pathologist reviewed 330 scanned NSCLC whole-slide images (WSIs) stained with the PD-L1 IHC 22C3 assay. From these, 49 cases were curated to cover the full TPS range and include cases with PD-L1 expression levels near clinically relevant thresholds. The selected sections were subsequently restained with the CD68 PG-M1 (GA613) marker and scanned again, to generate sequentially stained PD-L1 + CD68 WSI pairs. Both WSIs in each pair were globally aligned with RANSAC-optimized Euclidean transform, using ORB features (Rublee et al., 2011), followed by DIS optical flow (Kroeger et al., 2016) to enable pixel-accurate training, with the sequentially stained PD-L1 + CD68 WSI serving as a measurement-based ground truth.

To focus the model on clinically relevant areas the study pathologist annotated tumor regions on each sequentially stained WSI pair for use during model training. In addition, six particularly challenging regions (ROIs, Appendix E) were selected for cell-level evaluation based on the difficulty of distinguishing macrophages from tumor cells or the presence of macrophages infiltrating PD-L1-positive tumor areas, making them critical for assessing the efficacy of the proposed VSM method.

Our purpose was to study the efficacy of the VSM method rather than the performance of a specific model. Considering the challenging nature of the regions chosen for evaluation, we adopted a leave-tissue-out methodology. A separate model was trained for each evaluation region, excluding the entire WSI containing that region from the training data. This prevented any tissue-specific overfitting, while maximizing overall training set diversity.

The training hyperparameters were kept consistent across all models, using the same settings reported in the ablation tests section of (Ben-David et al., 2024), ensuring comparability across results.

As previously reported (Ben-David et al., 2024), inspection of model prediction by the study pathologist revealed the VSM patterns follow closely true sequential staining. However, to quantitatively evaluate macrophage detection performance, the study pathologist provided cell-level annotations for each of the six evaluation regions, marking cells as CD68-positive (CD68+) or CD68-negative (CD68-) based on the paired and aligned PD-L1 and sequentially stained PD-L1 + CD68 WSIs (Figure 2B). These annotations served as the ground-truth reference for model evaluation. With the aid of sequential CD68 staining, many cells could be confidently labeled by the study pathologist. However, some CD68 staining patterns were weak or could not be conclusively associated to a specific cell. Such cells were annotated as "unknown", some examples are presented in Appendix B.

Three additional certified pathologists, P1-P3, independently performed two rounds of cell-level annotation for each region. To ensure the same cells were annotated by all pathologists, the cells in each region were pre-marked, and set to an "unclassified" class. In the first round, they classified cells using only the PD-L1-stained WSI (Figure 2A). After a washout period of more than six months to minimize recall bias, they repeated the annotation task using pairs of PD-L1 and VSM CD68 WSIs (Figure 2C).

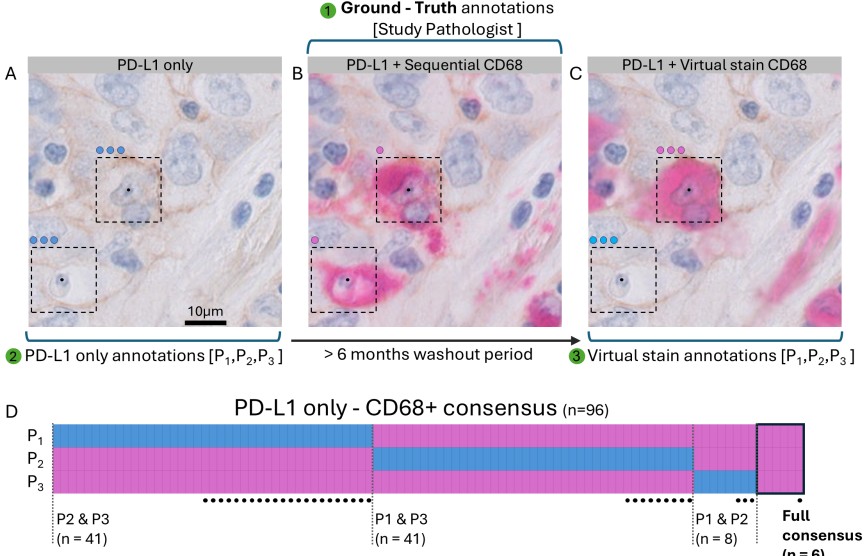

Figure 2: Cell-level annotations, represented by dots on top of squares marking sample cells, with CD68- and CD68+ classes colored blue and magenta, respectively. Annotations of (A) PD-L1-only WSIs and (C) PD-L1 + VSM CD68, by P1-P3 (B) Ground-truth by the study pathologist, using PD-L1 + CD68 IHC sequentially stained on the same section. Note full-consensus false-negative due to model error on bottom-left cell. (D) Agreement analysis of consensus annotated CD68+ cells using PD-L1-only WSIs. Black dots indicate cells with CD68+ ground truth annotations.

Since the task was defined as macrophage detection task, cells marked as "unknown" by P1-P3 were treated as CD68-, as the pathologists did not positively identify them as macrophages. To assess the robustness of this assumption, an alternative analysis was conducted (Appendix A) where unknown cells were instead treated as CD68+, yielding qualitatively similar results. To ensure that the reference used for evaluation was reliable, cells annotated as "unknown" by the study pathologist were excluded from macrophage detection performance metrics. Alternative analyses including these cells (set as either CD68+ or CD68-) are provided in Appendix A, presenting similar trends to the main-text results.

Inter-pathologist agreement was measured using Fleiss's Kappa for overall agreement across the three annotators and Cohen's Kappa for pairwise agreement between each pathol-

ogists pair (Fleiss and Cohen, 1973). Performance metrics were computed for each of P1-P3, comparing their annotations to the ground truth annotations by the study pathologist. Consensus labels were derived from majority-voting among P1-P3, full-consensus labels were obtained for cells where the annotations of P1-P3 were the same. Given the class imbalance present in the dataset, F1 scores were used as the primary performance measure, supplemented by precision, recall, and specificity (C).

## 3. Experimental Results

The analysis included a total of 2,304 annotated cells, of which 457 were marked as CD68 positive in the ground truth annotations provided by the study pathologist. Figure 2D summarizes the predictions of pathologists P1-P3 for a subset of 96 cells where the consensus PD-L1-only annotation indicated CD68 positivity. Among these 96 cells, only 35 were confirmed as macrophages in the ground truth annotations by the study pathologist, as indicated by dots in the figure. Notably, full consensus among all three pathologists was reached for only six cells, and of these, just one was confirmed as a macrophage by the ground truth; four cells were annotated as CD68-, and one classified as "unknown".

Table 1: Inter-pathologist agreement measured by Fleiss' and Cohen's kappa for PD-L1 and virtual staining multiplexing (VSM).

| Method | Fleiss' | Cohen's | | |
|---|---|---|---|---|
| | P1 vs. P2 vs. P3 | P1 vs. P2 | P1 vs. P3 | P2 vs. P3 |
| PD-L1 | -0.11 | 0.07 | -0.03 | 0.01 |
| VSM | 0.62 | 0.79 | 0.51 | 0.58 |

Table 1 presents Fleiss's and Cohen's kappa measures of inter-rater agreement for P1-P3, both with and without the assistance of VSM CD68. Agreement was significantly improved with the aid of VSM: Fleiss' kappa increased from -0.1 (indicating extremely poor agreement) with PD-L1-only annotations to 0.62 (substantial agreement) with VSM.

The number of cells with full consensus annotations as CD68+ also increased markedly, rising to 463 with the use of VSM. Even the pair of P1 and P2, who demonstrated the highest agreement among the pairings, showed only slight agreement with PD-L1-only annotations (Cohen's kappa = 0.07). With VSM, all pairwise agreements improved to at least moderate levels (Cohen's kappa > 0.5).

The effectiveness of pathologists in detecting macrophages was further analyzed using VSM CD68. Figure 3 provides detailed comparisons of performance with and without VSM. Figure 3A–C present receiver operating characteristic (ROC) plots for P1-P3, displaying recall and 1-specificity across six evaluation regions.

Without the assistance of VSM, P1 and P2 exhibited low recall (0.06, 0.08) and high specificity (0.94, 0.97) across all evaluation regions, while P3 achieved higher recall (0.59) but at the expense of reduced specificity (0.66). The introduction of VSM substantially enhanced recall for P1 and P2 (0.58, 0.65), albeit with some decreases in specificity (0.9,

0.87). For P3, both recall (0.75) and specificity (0.76) improved across all evaluation regions with the aid of VSM.

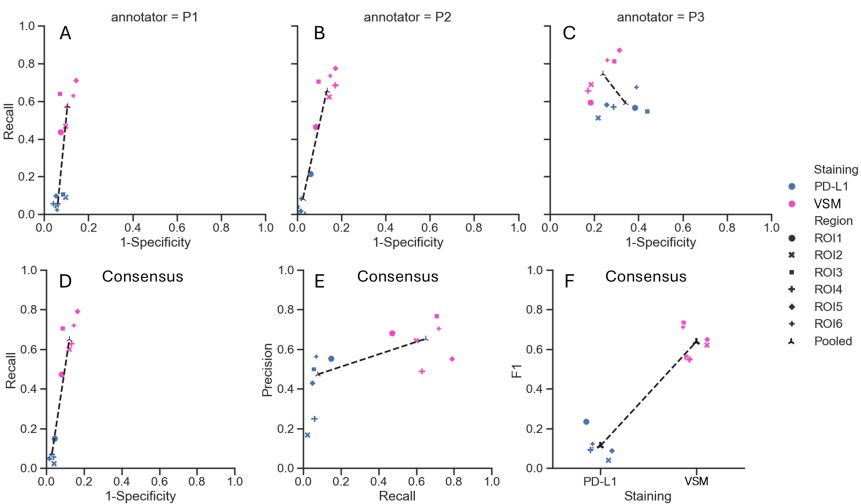

Figure 3: Performance metrics compared to the study pathologist ground-truth across six evaluation regions with and without assistive virtual staining multiplexing. Changes in pooled performance metrics are indicated by a dashed black line. (A-C) Receiver operating characteristic (ROC) plots for P1-P3, respectively. (D) ROC plot, (E) Precision-recall plot and (F) F1 scores for majority-vote consensus annotations.

Figure 3D–F focuses on the performance of consensus annotations. Figure 3D illustrates the ROC plot for consensus annotations, showing a similar trend to the individual performances of P1 and P2, with low recall (0.077) and high specificity (0.97) in the absence of VSM. VSM led to much higher recall (0.65), accompanied by a decrease in specificity (0.88). Figure 3E depicts precision-recall curves, highlighting notably improved pooled recall (0.65) and precision (0.65), with precision increases of 2–3 times for certain regions. Finally, Figure 3F summarizes F1 scores, which consistently improved with VSM. The pooled F1 score for consensus annotations showed a five-fold improvement with the aid of VSM, rising from 0.13 to 0.65.

Comparing majority-vote consensus annotations to full-consensus annotations revealed additional insights. Without VSM, full-consensus performance was notably low, with an F1 score of 0.012, recall of 0.006, and precision of 0.2. VSM significantly improved these metrics, raising them to 0.71, 0.72, and 0.69, respectively. Appendix C provides a comprehensive table of evaluation metrics for all annotators and regions, with and without the use of VSM.

## 4. Conclusion

In this study, we presented a comprehensive evaluation of VSM for CD68 to assist pathologists in detecting macrophages within the tumor microenvironment of PD-L1-stained NSCLC slides.

Unassisted macrophage detection showed agreement levels even lower than those observed in clinical scoring tasks such as HER2 and PD-L1, where inter-pathologist agreement is already below ideal thresholds(Robbins et al., 2023; Han et al., 2022). Unlike such scoring tasks, where pathologists' annotations often serve as the only ground truth, our study leveraged sequentially stained CD68 IHC to provide an independent and more objective ground truth. Comparisons to this ground truth revealed consistently poor unassisted performance across the curated challenging regions.

The analysis of individual pathologists' annotations further revealed notable variability. Two pathologists exhibited similar overall performance metrics, while the third displayed distinct tendencies, potentially reflecting differences in training or decision-making styles (see Appendix D for annotation distributions). However, even the two pathologists with similar overall metrics showed low agreement on a per-cell basis, as evidenced by their low Cohen's kappa values. These results suggest that individual variability remains a challenge, even among pathologists with ostensibly similar capabilities.

Macrophage detection is explicitly required in clinical TPS assessments to exclude macrophages from scoring. The poor unassisted performance observed here suggests that macrophage misclassification could affect TPS scoring in challenging cases, underscoring the need for improved tools to assist in these scenarios.

The introduction of VSM CD68 markedly improved pathologists' performance and agreement. Detection errors with VSM conflate errors due to model imperfections and annotator mistakes. The reported full-consensus performance with VSM may be considered a proxy measure to model errors, in concordance with anecdotal fault analysis of model predictions (e.g. Figure 2C), where missing VSM CD68 staining led to false negative full-consensus classification. These results indicate that further refinement of the model would be beneficial to reduce such errors. Additionally, with VSM, the annotation statistics of all pathologists became more closely aligned, demonstrating a reduction in inter-pathologist variability.

Consensus annotations provide further insight into the difficulty of macrophage detection. Without VSM, majority-vote consensus annotations failed to mitigate the challenges of unassisted detection. Full-consensus annotations performed even worse, with only six macrophages identified across the regions. However, both majority-vote and full-consensus annotations improved substantially with VSM.

Our findings suggest that tasks with poor agreement among pathologists, such as unassisted macrophage detection, may pose a challenge for the development of AI-based diagnostic tools. Manual annotations from individual pathologists may not provide reliable ground truth in these contexts. Prior studies (Robbins et al., 2023; Han et al., 2022) have shown that requiring agreement among multiple pathologists might reduce the dataset size significantly. This aligns with our observation that without the assistance of VSM the number of full-consensus annotations was drastically limited. Furthermore, when compared to sequential-stain based ground truth, full consensus was found to perform worse than majority vote for unassisted macrophage detection. These results suggest that increasing the

number of annotators or requiring unanimity does not address the fundamental challenges of low-agreement tasks.

The scarcity of reliably annotated medical data poses additional challenges for method evaluation. To mitigate overfitting, we employed a leave-tissue-out strategy, training separate models for each tissue in our evaluation set. We note that comparison with our previously reported model (Ben-David et al., 2024), which likely exhibited overfitting due to within-tissue evaluation, reveals a decrease in performance metrics with the current approach. However, the qualitative conclusions remained consistent, indicating that within-tissue evaluation may be sufficient for exploratory, proof-of-concept studies, particularly when time or computational resources are limited.

Inclusion of VSM in pathology workflows is computationally feasible. An NVIDIA RTX A5000 Laptop GPU can virtually stain an X40 field of view (250um radius) in under a second, enabling real-time use, while an NVIDIA Tesla V100 GPU achieves the same in 0.5 seconds, supporting efficient parallelized offline processing of WSIs.

This in-depth evaluation demonstrates that VSM CD68 significantly improves pathologists' ability to detect macrophages. This capability is particularly important for borderline PD-L1 slide scoring and may offer opportunities for spatial analysis of macrophages in the tumor microenvironment. Additionally, our work demonstrates that VS methods trained using measurement-based ground truth, can effectively address the challenges of low inter-pathologist agreement and the need for extensive ground-truth data, enabling robust performance with minimal reliance on manual annotation. Further work would be needed to further develop and validate on a larger, more diverse dataset that includes sections stained and scanned by different labs.

Regulatory statement: This study is for proof of concept only. This paper does not imply any clinical functionality.

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

## Appendix A. Alternative Analysis

This section contains analysis results with alternative choices for the handling of the "unknown" classes. Table 2 and Figure 4 present results where "unknown" annotations by P1-P3 were set to CD68+.

Table 2: Alternative analysis of inter-pathologist agreement, with "unknown" cells set to CD68+, measured by Fleiss' and Cohen's kappa for PD-L1 and virtual stain multiplexing (VSM).

| Method | Fleiss' | Cohen's | | |
|---|---|---|---|---|
| | P1 vs. P2 vs. P3 | P1 vs. P2 | P1 vs. P3 | P2 vs. P3 |
| PD-L1 | 0.05 | 0.14 | 0.23 | 0.03 |
| VSM | 0.54 | 0.60 | 0.56 | 0.48 |

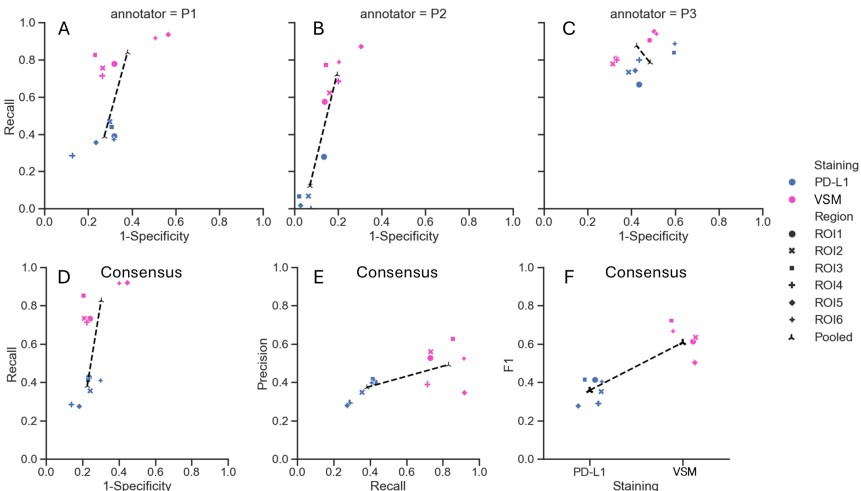

Figure 4: Performance metrics with P1-P3 "unknown" annotations set to CD68+ (A-C) Receiver operating characteristic (ROC) plots for P1-P3. (D) ROC plot for consensus annotations. (E) Precision-recall plot for consensus annotations. (F) F1 scores for consensus annotations.

Figure 5 and Figure 6 present performance metrics where the study pathologist's "unknown" annotations were not excluded from the analysis. Instead, both the study pathologist's "unknown" annotations and P1-P3 "unknown" annotations were set to CD68- or CD68+, respectively.

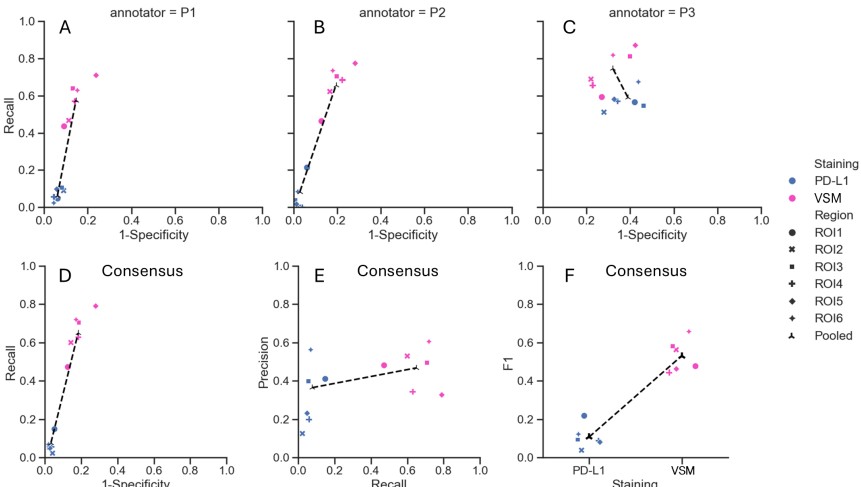

Figure 5: Performance metrics with all "unknown" annotations set to CD68-, the study pathologist's "unknown" annotated cells not omitted from analysis as in main text. (A-C) Receiver operating characteristic (ROC) plots for P1-P3. (D) ROC plot for consensus annotations. (E) Precision-recall plot for consensus annotations. (F) F1 scores for consensus annotations.

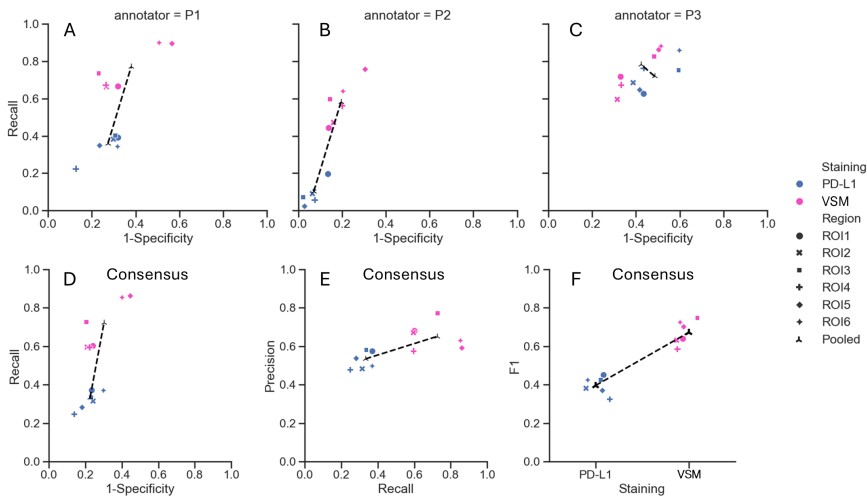

Figure 6: Performance metrics with all "unknown" annotations set to CD68+, the study pathologist's "unknown" annotated cells not omitted from analysis as in main text. (A-C) Receiver operating characteristic (ROC) plots for P1-P3. (D) ROC plot for consensus annotations. (E) Precision-recall plot for consensus annotations. (F) F1 scores for consensus annotations.

## Appendix B.  Data Annotation Workflow

### B.1.  Generating ground-truth

Preprocessing: an automatic nuclei detection algorithm was applied to each evaluation region before annotation, with all detected nuclei initially assigned to an unclassified class. The study pathologist reviewed the nuclei detection results and adjusted them as needed by adding or removing nuclei markers.

Expert Annotation: The study pathologist classified each detected nucleus using an in-house whole-slide image viewer. The classification was performed while simultaneously examining the PD-L1 stained slide alongside the sequentially stained PD-L1 +CD68 IHC slide in a side-by-side arrangement.

- Nuclei surrounded by strong magenta staining were classified as macrophages (CD68+).

- An additional "unknown" class was introduced to categorize cells where the nearby magenta staining was weak or difficult to associate with a specific nucleus.

- Other cells were classified as non-macrophages (CD68-)

- See examples in Figure 7

### B.2.  Inter-Observer Annotation

Pathologist Enrollment: Participating pathologists were certified in scoring PD-L1 slides.

#### B.2.1. Phase 1 – PD-L1 Slides Only

Preprocessing: Before each annotation phase (with or without the virtual stain marker (VSM) for CD68), all detected nuclei were reset to the unclassified class.

Training and Instructions: Upon enrollment, pathologists received training on using the in-house annotation software.

Annotation Process: Pathologists were provided with only PD-L1 stained slides and were instructed to annotate based on their prior knowledge and familiarity with PD-L1 staining.

#### B.2.2. Phase 2 – PD-L1 + VSM CD68

Timing: This phase was conducted after a six-month washout period.

Preprocessing: All detected nuclei were reset to the unclassified class, and an additional VSM layer for CD68 was introduced.

Training and Instructions: The study pathologist provided an introduction to the characteristics of CD68 staining and instructed annotators on how to classify cells into CD68+, CD68-, or Unknown categories.

Annotation Process: Annotations were performed using a side-by-side arrangement of the PD-L1 and PD-L1 + VSM CD68 slides.

Note, all annotations tasks were done without time limits

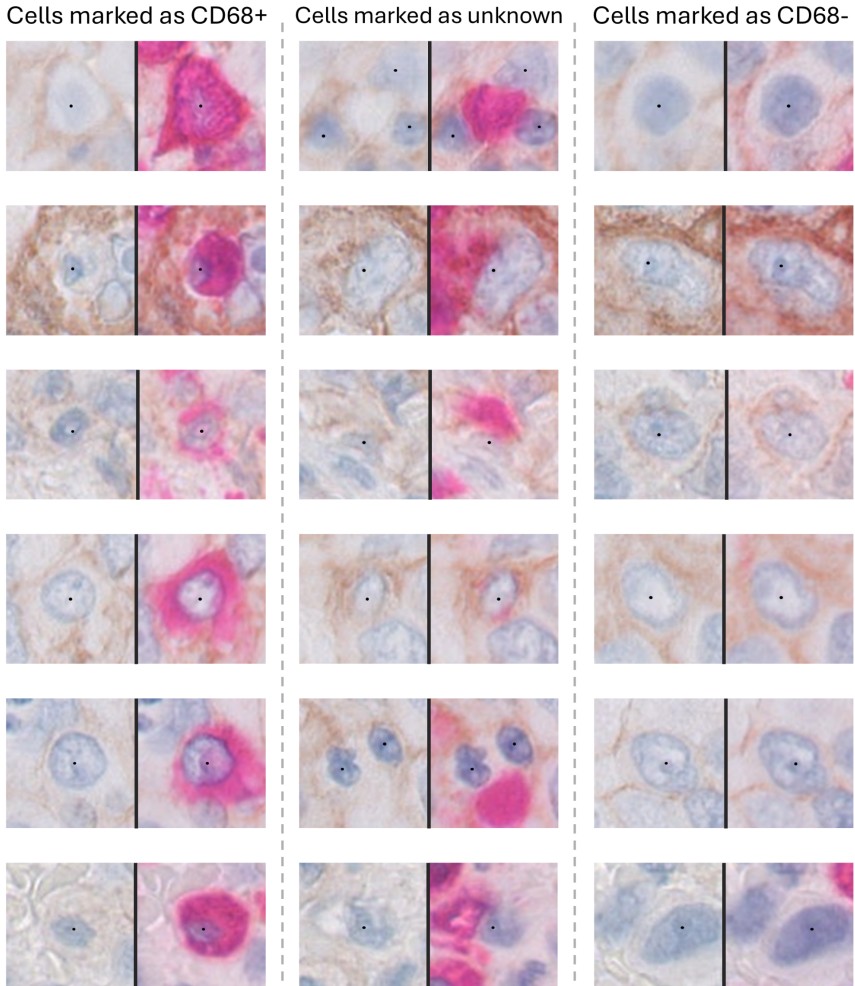

Figure 7: Ground Truth Annotation. Image pairs of cells stained with PD-L1 only and PD-L1 + sequential CD68 IHC. (Left) cells classified as CD68+, where magenta staining is clearly associated with the cells marked by black dots. (Center) cells classified as "unknown" by the study pathologist, where the association between nearby staining and the marked cells (black dots) was inconclusive. (Right) cells classified as CD68-, as no magenta staining is observed in association with the marked cells.

## Appendix C. Performance Metrics

Performance metrics for macrophage detection were computed for each of P1-P3 by comparing their annotations to the ground-truth annotations of the study pathologist, both with and without the assistance of VSM CD68. The possible classes for evaluation were positive (CD68+) and negative (CD68-). Given the class imbalance present in the dataset, F1 scores were used as the primary performance measure, supplemented by precision, recall, and specificity. Below are the definitions we used in computing these metrics:

$$F1 = \frac{2}{recall^{-1} + precision^{-1}} \tag{1}$$

$$Precision = \frac{TP}{TP + FP} \tag{2}$$

$$Recall = \frac{TP}{TP + FN} \tag{3}$$

$$Specificity = \frac{TN}{TN + FP} \tag{4}$$

Where TP, TN, FP and FN refer to true positives, true negatives, false positives and false negatives respectively. Metrics were computed for each evaluation region independently and pooled metrics were computed by pooling all annotations from all regions and computing the metrics using the overall TP, TN, FP and FN.

The below tables present pooled performance metrics as well as metrics for each region separately. Each table contains metrics for majority-vote consensus annotation and each of P1-P3, with and without the assistance of virtual stain multiplexing.

Table 3: Pooled performance metrics

| annotator | Staining | Region | Precision | Recall | F1 | Specificity |
|---|---|---|---|---|---|---|
| Consensus | PD-L1 | Pooled | 0.47 | 0.08 | 0.13 | 0.97 |
| Consensus | VSM | Pooled | 0.65 | 0.65 | 0.65 | 0.88 |
| P1 | PD-L1 | Pooled | 0.26 | 0.06 | 0.1 | 0.94 |
| P1 | VSM | Pooled | 0.66 | 0.58 | 0.61 | 0.9 |
| P2 | PD-L1 | Pooled | 0.53 | 0.08 | 0.14 | 0.97 |
| P2 | VSM | Pooled | 0.63 | 0.66 | 0.64 | 0.87 |
| P3 | PD-L1 | Pooled | 0.38 | 0.59 | 0.46 | 0.66 |
| P3 | VSM | Pooled | 0.53 | 0.75 | 0.62 | 0.76 |

Table 4: Performance metrics for ROI1

| annotator | Staining | Region | Precision | Recall | F1 | Specificity |
|---|---|---|---|---|---|---|
| Consensus | PD-L1 | ROI1 | 0.55 | 0.15 | 0.23 | 0.96 |
| Consensus | VSM | ROI1 | 0.68 | 0.47 | 0.56 | 0.92 |
| P1 | PD-L1 | ROI1 | 0.24 | 0.05 | 0.08 | 0.95 |
| P1 | VSM | ROI1 | 0.68 | 0.44 | 0.53 | 0.93 |
| P2 | PD-L1 | ROI1 | 0.56 | 0.21 | 0.31 | 0.94 |
| P2 | VSM | ROI1 | 0.67 | 0.46 | 0.55 | 0.91 |
| P3 | PD-L1 | ROI1 | 0.35 | 0.56 | 0.43 | 0.62 |
| P3 | VSM | ROI1 | 0.54 | 0.59 | 0.57 | 0.82 |

Table 5: Performance metrics for ROI2

| annotator | Staining | Region | Precision | Recall | F1 | Specificity |
|---|---|---|---|---|---|---|
| Consensus | PD-L1 | ROI2 | 0.17 | 0.02 | 0.04 | 0.96 |
| Consensus | VSM | ROI2 | 0.64 | 0.6 | 0.62 | 0.88 |
| P1 | PD-L1 | ROI2 | 0.25 | 0.09 | 0.13 | 0.9 |
| P1 | VSM | ROI2 | 0.64 | 0.47 | 0.54 | 0.9 |
| P2 | PD-L1 | ROI2 | 0.0 | 0.0 | 0.0 | 0.99 |
| P2 | VSM | ROI2 | 0.61 | 0.62 | 0.62 | 0.85 |
| P3 | PD-L1 | ROI2 | 0.46 | 0.51 | 0.48 | 0.78 |
| P3 | VSM | ROI2 | 0.57 | 0.69 | 0.63 | 0.81 |

Table 6: Performance metrics for ROI3

| annotator | Staining | Region | Precision | Recall | F1 | Specificity |
|---|---|---|---|---|---|---|
| Consensus | PD-L1 | ROI3 | 0.5 | 0.05 | 0.1 | 0.98 |
| Consensus | VSM | ROI3 | 0.77 | 0.71 | 0.74 | 0.91 |
| P1 | PD-L1 | ROI3 | 0.33 | 0.11 | 0.16 | 0.91 |
| P1 | VSM | ROI3 | 0.79 | 0.64 | 0.71 | 0.93 |
| P2 | PD-L1 | ROI3 | 1.0 | 0.04 | 0.08 | 1.0 |
| P2 | VSM | ROI3 | 0.75 | 0.71 | 0.73 | 0.9 |
| P3 | PD-L1 | ROI3 | 0.33 | 0.55 | 0.41 | 0.56 |
| P3 | VSM | ROI3 | 0.53 | 0.81 | 0.64 | 0.71 |

Table 7: Performance metrics for ROI4

| annotator | Staining | Region | Precision | Recall | F1 | Specificity |
|---|---|---|---|---|---|---|
| Consensus | PD-L1 | ROI4 | 0.25 | 0.06 | 0.09 | 0.97 |
| Consensus | VSM | ROI4 | 0.49 | 0.63 | 0.55 | 0.87 |
| P1 | PD-L1 | ROI4 | 0.22 | 0.06 | 0.09 | 0.96 |
| P1 | VSM | ROI4 | 0.53 | 0.57 | 0.55 | 0.9 |
| P2 | PD-L1 | ROI4 | 0.0 | 0.0 | 0.0 | 0.97 |
| P2 | VSM | ROI4 | 0.44 | 0.69 | 0.54 | 0.83 |
| P3 | PD-L1 | ROI4 | 0.29 | 0.57 | 0.38 | 0.71 |
| P3 | VSM | ROI4 | 0.43 | 0.66 | 0.52 | 0.83 |

Table 8: Performance metrics for ROI5

| annotator | Staining | Region | Precision | Recall | F1 | Specificity |
|---|---|---|---|---|---|---|
| Consensus | PD-L1 | ROI5 | 0.43 | 0.05 | 0.09 | 0.98 |
| Consensus | VSM | ROI5 | 0.55 | 0.79 | 0.65 | 0.83 |
| P1 | PD-L1 | ROI5 | 0.32 | 0.1 | 0.15 | 0.95 |
| P1 | VSM | ROI5 | 0.56 | 0.71 | 0.62 | 0.86 |
| P2 | PD-L1 | ROI5 | 0.2 | 0.02 | 0.03 | 0.98 |
| P2 | VSM | ROI5 | 0.53 | 0.77 | 0.63 | 0.83 |
| P3 | PD-L1 | ROI5 | 0.37 | 0.58 | 0.45 | 0.74 |
| P3 | VSM | ROI5 | 0.42 | 0.87 | 0.56 | 0.69 |

Table 9: Performance metrics for ROI6

| annotator | Staining | Region | Precision | Recall | F1 | Specificity |
|---|---|---|---|---|---|---|
| Consensus | PD-L1 | ROI6 | 0.56 | 0.07 | 0.12 | 0.97 |
| Consensus | VSM | ROI6 | 0.7 | 0.72 | 0.71 | 0.85 |
| P1 | PD-L1 | ROI6 | 0.16 | 0.02 | 0.04 | 0.94 |
| P1 | VSM | ROI6 | 0.7 | 0.63 | 0.66 | 0.87 |
| P2 | PD-L1 | ROI6 | 0.69 | 0.08 | 0.15 | 0.98 |
| P2 | VSM | ROI6 | 0.7 | 0.73 | 0.72 | 0.85 |
| P3 | PD-L1 | ROI6 | 0.45 | 0.67 | 0.54 | 0.61 |
| P3 | VSM | ROI6 | 0.6 | 0.82 | 0.69 | 0.74 |

## Appendix D. Annotation Distributions

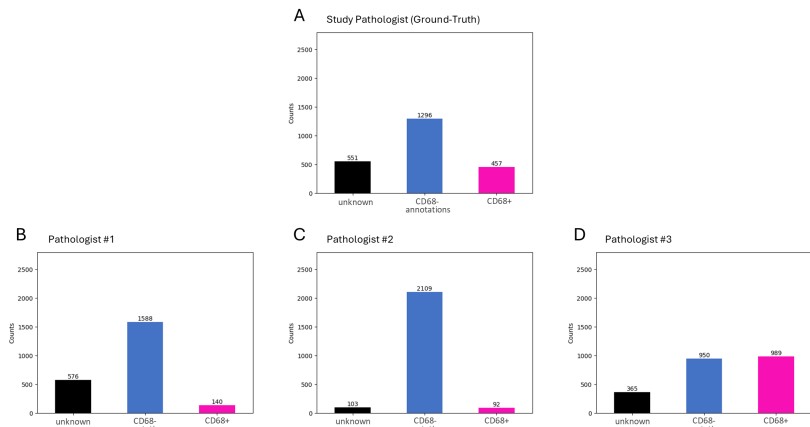

Figure 8: Annotation Distributions. Panel A presents annotations distributions for the study pathologist, using both PD-L1 and PD-L1 + sequential CD68 staining (used as ground-truth annotations for performance analysis). Distributions of P1–P3 annotations, using PD-L1 staining only, for the same cells annotated by the study pathologist, given in panels B-D, reveal notable variability. Pathologist P1 and P2 exhibit somewhat similar annotation distributions, seen in panels B and C, respectively (although agreeing on only 14 cells). Pathologist P3 annotation distribution given in panel D is starkly different, potentially reflecting differences in training or decision-making styles. We note that most of the cells marked as CD68+ by P1-P3 are false positives.

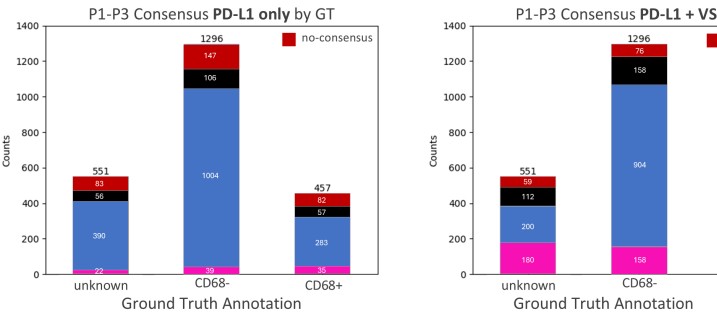

Figure 9: Consensus Annotation Distributions. Stacked majority-vote consensus annotations by the study pathologist's ground-truth annotations. (left) P1-P3 consensus annotations using only PD-L1 stained slides. (right) P1-P3 consensus annotations with the addition of VSM CD68. The increase in overall CD68+ annotations by P1-P3 when assisted by VSM is apparent. In addition, the number of cells where no consensus was attained decreased with the addition of VSM.

## Appendix E. Evaluation Regions

The below figures present the ROIs used for macrophage detection evaluation. For each ROI, we give the PD-L1 only, PD-L1 + VSM CD68 and PD-L1 + sequential CD68 IHC.

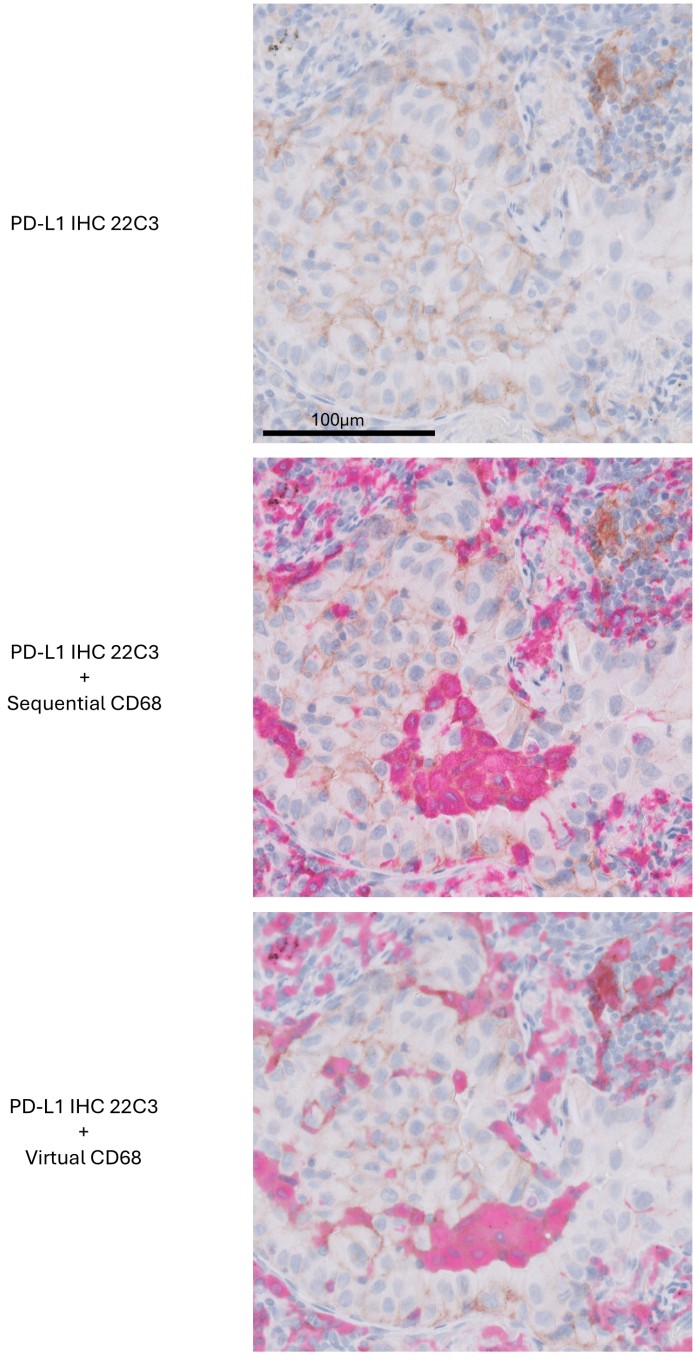

Figure 10: ROI1

PD-L1 IHC 22C3

PD-L1 IHC 22C3
+
Sequential CD68

PD-L1 IHC 22C3
+
Virtual CD68

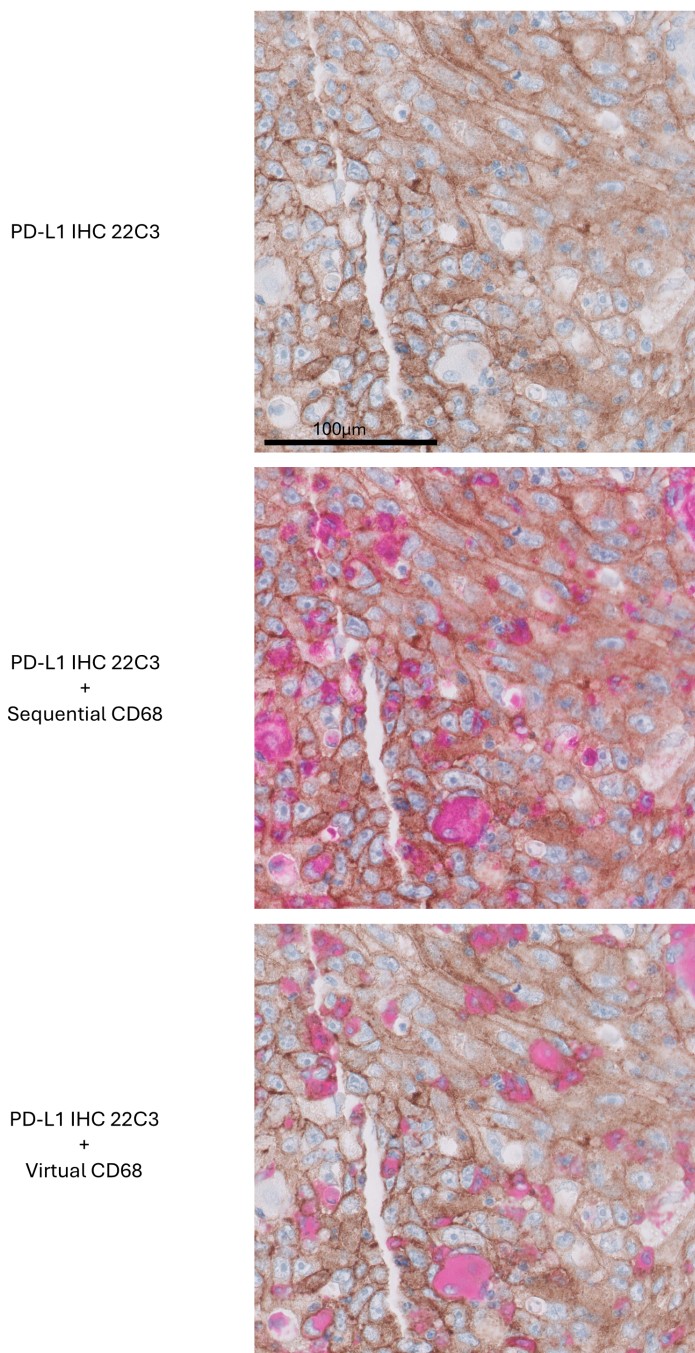

Figure 11: ROI2

PD-L1 IHC 22C3

PD-L1 IHC 22C3
+
Sequential CD68

PD-L1 IHC 22C3
+
Virtual CD68

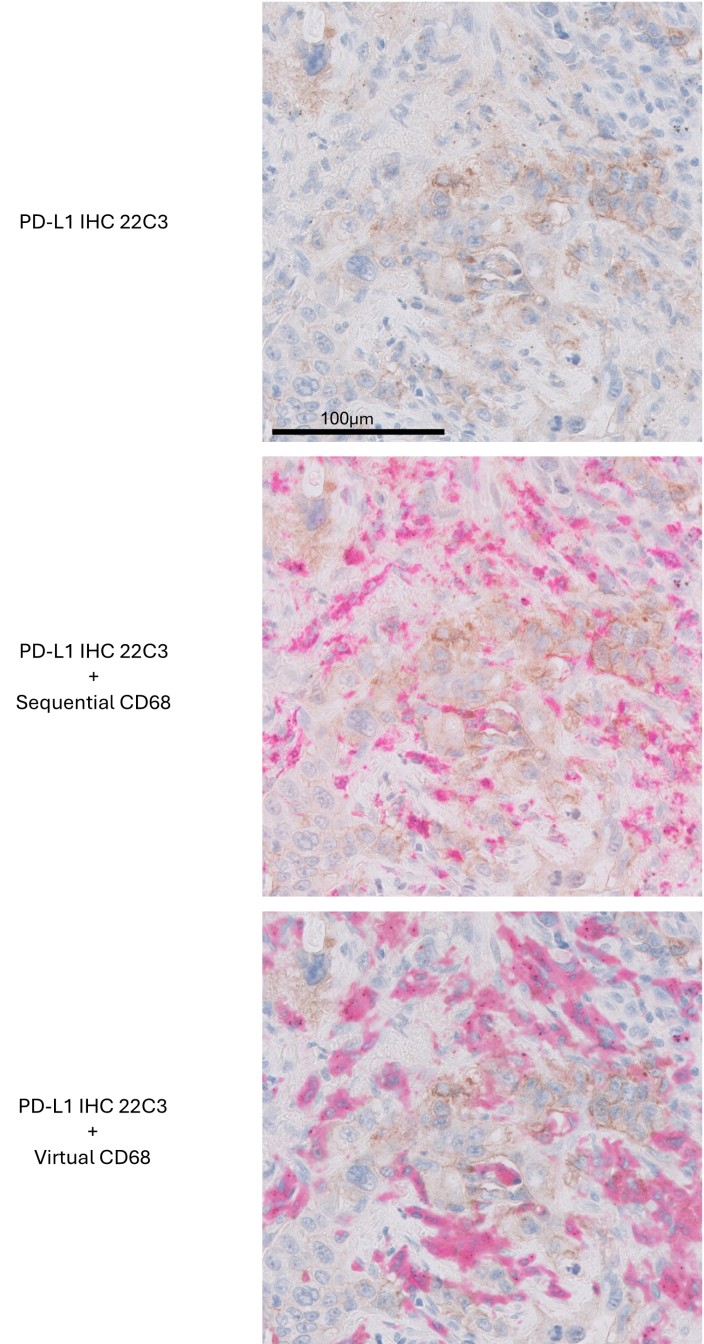

Figure 12: ROI3

PD-L1 IHC 22C3

PD-L1 IHC 22C3
+
Sequential CD68

PD-L1 IHC 22C3
+
Virtual CD68

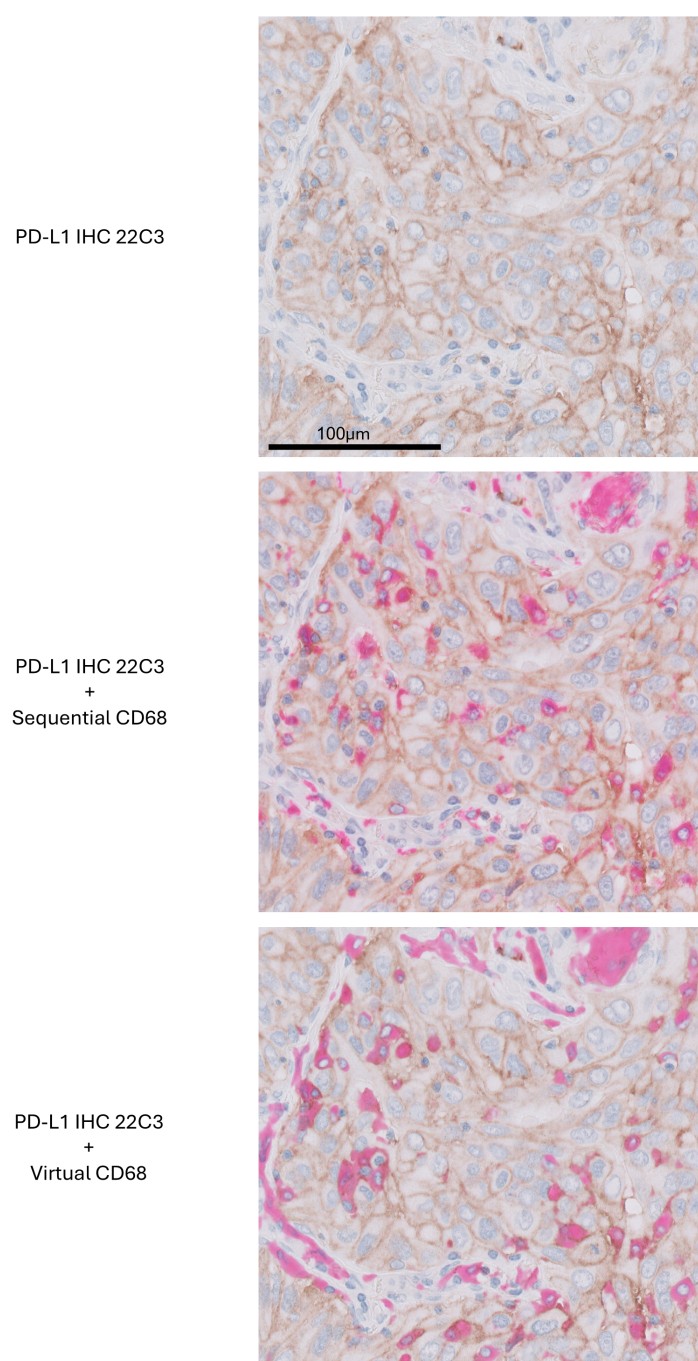

Figure 13: ROI4

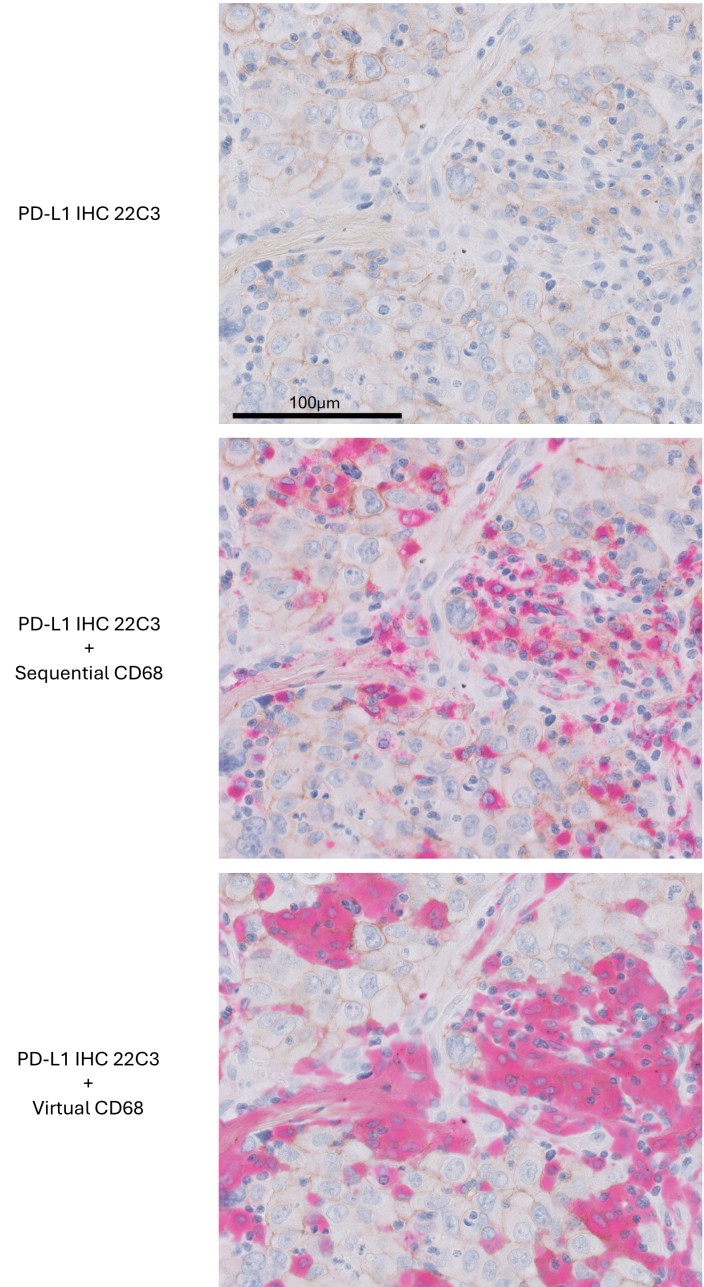

Figure 14: ROI5

PD-L1 IHC 22C3

PD-L1 IHC 22C3
+
Sequential CD68

PD-L1 IHC 22C3
+
Virtual CD68

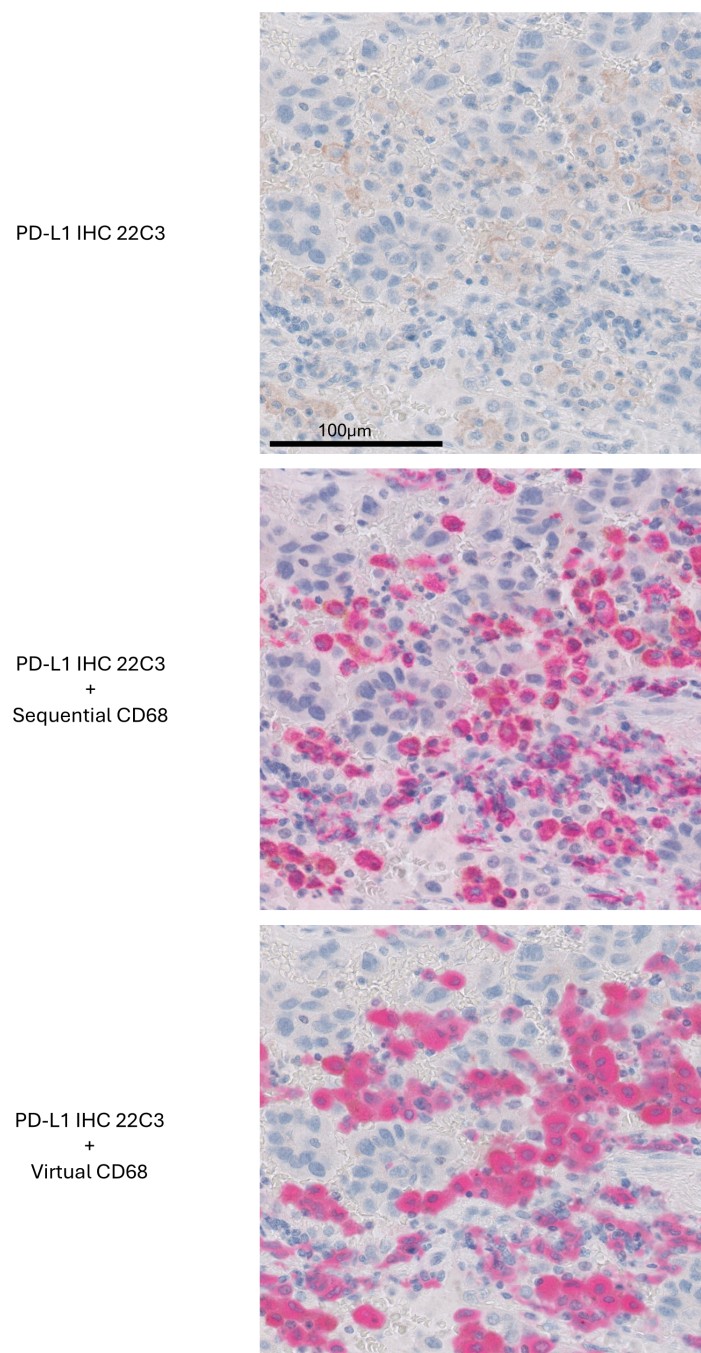

Figure 15: ROI6

