# OpenReview forum: "Evaluation of Virtual Stain Multiplexed CD68 for Macrophage Detection in NSCLC PD-L1 Slides"
_MIDL.io/2025/Conference — MIDL 2025 Poster_

### Official Review · Reviewer_hZgE · 2025-02-20

**Confidence:** 4
**Preliminary Rating:** 3
**Recommendation:** Poster
**Final Rating:** 4

**Summary:**

This paper presents an in-depth evaluation of a virtual stain multiplexing (VSM) approach for enhancing macrophage detection in NSCLC PD-L1 immunohistochemistry (IHC) slides. By leveraging a U-Net–based model that simulates sequential CD68 staining via the Beer-Lambert law, the authors generate a virtual CD68 stain from PD-L1–stained whole slide images (WSIs). The method is designed to address common challenges in manual slide interpretation, such as inter-observer variability and limited tissue availability, by providing measurement-based ground truth for training. Quantitative results demonstrate substantial improvements in inter-pathologist agreement—with Fleiss’ kappa rising from –0.1 to 0.62—and overall detection performance, as evidenced by an increase in F1 score from 0.13 to 0.65, underscoring the potential of VSM to improve diagnostic consistency in pathology.

**Strengths:**

The paper tackles a critical and practical challenge in digital pathology by proposing a VSM method that virtually simulates additional staining, thereby reducing reliance on costly and labor-intensive physical stains. One major strength is the rigorous evaluation of the approach: the authors conducted detailed cell-level annotation studies involving multiple pathologists and compared performance both with and without VSM assistance. The substantial improvement in inter-observer agreement, as measured by both Fleiss’ and Cohen’s kappa, along with significant gains in recall and F1 score, highlights the clinical relevance of the work. Additionally, the method is underpinned by a clear physical model based on the Beer-Lambert law and is implemented using a U-Net architecture, which facilitates reproducibility and potential integration into existing digital pathology workflows. The comprehensive experimental design, including leave-tissue-out validation and alternative analyses, further reinforces the robustness of the findings.

**Weaknesses:**

1. One of my major concern is the relatively small dataset size (49 curated WSIs), which might limit the generalizability of the findings across diverse clinical scenarios and tissue types. The study focuses exclusively on NSCLC PD-L1 slides and macrophage detection; thus, it remains unclear how well the proposed VSM approach would perform for other stains or in different diagnostic contexts.
2. The reliance on sequentially stained slides as measurement-based ground truth introduces potential alignment errors and may not fully capture the variability encountered in routine clinical practice. Can the authors discuss the limitation/challenges it bring by this slice-to-slice difference.
3. Since this paper is less methodological but more towards clinical evaluation of an established method, the paper could also benefit from a more detailed discussion of computational efficiency and potential integration challenges in a real-world diagnostic pipeline, as well as a comparison with alternative virtual staining or multiplexing techniques currently emerging in the literature.

**Detailed Comments:**

The paper is well-organized and provides a comprehensive evaluation of a novel VSM approach for macrophage detection. The use of a U-Net architecture, combined with transformation into optical density space based on the Beer-Lambert law, is a particularly innovative aspect that lends physical interpretability to the virtual staining process. The experimental section is robust, with detailed inter-observer agreement studies that clearly demonstrate the improvement in pathologist performance when VSM is employed. The presentation of performance metrics—ROC curves, precision-recall plots, and F1 scores—across multiple challenging regions provides convincing evidence of the method's efficacy. However, the study is limited by the relatively small number of WSIs and the focus on a single cancer type and stain. A discussion on the scalability of the approach, as well as its applicability to other types of stains or tissue, would enhance the manuscript. Additionally, insights into potential misclassification cases and failure modes could further clarify the boundaries of the method’s effectiveness.

**Justification Of The Final Rating:**

The authors’ rebuttal effectively addresses my concerns raised. They clarified i the rebuttal that the 49 WSIs were carefully selected from a larger pool to represent the full clinical spectrum. They also emphasized that sequential staining on the same tissue section eliminates slice-to-slice variability in the revised manuscript which clears the confusion. They further demonstrated that the method is computationally efficient and suitable for real-time clinical application using affordable GPUs, while also providing robust alternative analyses for handling ambiguous cases. Overall, the rebuttal convincingly supports the paper's contributions despite its limited focused scope. Thus, I am leaning towards a weak accept.

**Justification Of The Preliminary Rating:**

The paper makes a significant contribution by addressing the challenge of low inter-observer agreement in macrophage detection through a well-conceived virtual staining method. The experimental results, particularly the marked improvements in kappa statistics and F1 scores, are compelling and underscore the potential clinical utility of the approach. However, limitations regarding dataset size, generalizability to other stains or cancers, and integration challenges prevent a stronger recommendation. Further analysis on scalability and robustness across diverse clinical scenarios, as well as a deeper exploration of computational considerations, would be necessary to elevate the impact of the work. Overall, the contribution is valuable and innovative, justifying a weak accept with room for further refinement.

**Questions To Address In The Rebuttal:**

1. How does the VSM method perform when applied to a larger and more diverse dataset, possibly including different cancer types or stains?
2. Can the authors elaborate on the potential challenges in integrating this virtual staining approach into existing clinical workflows, particularly regarding computational efficiency and alignment accuracy?
3. What measures are in place to handle alignment errors between sequentially stained slides, and how might these errors affect the overall performance?
4. Could the authors provide additional insights into how the method handles ambiguous cases, particularly those cells annotated as "unknown," and the impact of different annotation handling strategies on the performance metrics?

---

> ### Author Response · Authors · 2025-03-08
>
> We thank the reviewer for your detailed and insightful summary of our study. We appreciate your recognition of our approach, methodology, and the impact of virtual stain multiplexing on improving inter-pathologist agreement and detection performance. Below, we provide our responses to the key points raised.
>
> **Weaknesses:**
> 1. We appreciate the reviewer’s concern regarding dataset size and generalizability. The study is focused on NSCLC and virtual stain multiplexing of PD-L1 and CD68 only, which is known to solve a particularly challenging task. Applications to other tissue types will require further development. While we recognize that validation on datasets from different labs and scanners would be valuable, that was beyond the scope of our current research. We added a statement to the Conclusion section to that effect.
> In addition, we have updated the manuscript to expand on the dataset curation process leading to the curated dataset comprising 49 WSIs. The 49 selected cases are not a random dataset. Instead this is a distillation of a 330 cases from which cases were selected to ensure that cases with TPS score covering the possible range are represented, as well as cases near the clinical threshold.
> 2. We apologize for not being clear on this point. To clarify, our sequential staining approach is performed on the same tissue section, ensuring that there is no slice-to-slice variability, which is a true concern in studies using adjacent sections. We have emphasized this point multiple times in the revised manuscript to avoid confusion.
> Additionally, we acknowledge that precise alignment is crucial for generating accurate measurement-based ground truth. To address this, we have expanded the description of our alignment process in the Methods section.
> 3. We recognize the importance of computational efficiency for practical implementation. Given its processing speed, this method could be applied in real-time using relatively unexpensive GPUS, making it feasible for integration into pathology workflows. Further acceleration possible using high-performance hardware. To clarify this, we have included additional details on processing times in the Conclusion section [p.9]. As stated, this study is proof of concept and further research is required for the efficient integration into the real-world pipeline.
>
> * A comprehensive comparison with alternative virtual staining and multiplexing techniques is provided in our previous publication (Ben-David et al., 2024). Our approach is unique in that, unlike other virtual staining methods, it adds (multiplexes) information on top of an existing IHC stain (which can potentially be applied on existing scans) rather than replacing into another staining.
>
> **Key Detailed Comments:**
> - Understanding misclassification cases and failure modes is essential for further improving and refining the model. To address this, we have clarified in the manuscript how detection errors with VSM may result from both model imperfections and annotator mistakes. We clarify this distinction more explicitly in the manuscript and suggested that full-consensus annotations analysis may serve as a reasonable proxy for assessing the model performance. We highlight an anecdotal false-positive case in Figure 2C, where a full-consensus annotation of a cell as CD68- is attributed to model fault, as actual magenta staining is visible in the sequential IHC staining (Figure 2B).
>   - We have clarified this observation explicitly in the Conclusion section [p.8]: “Detection errors with VSM conflate errors due to model imperfections and annotator mistakes. The reported full-consensus performance with VSM may be considered a proxy measure for model errors, aligning with anecdotal fault analysis of model predictions (e.g., Figure 2), where missing VSM CD68 staining led to false-negative full-consensus classification.”
>   - Additionally, we added a note in the Figure 2 caption [p.5]: “Note full-consensus false-negative due to model error on the bottom-left cell.”
>
> **Rebuttal Comments:**
> 1. Please see our responses to weakness comment 1.
> 2. & 3. Please see our response to weakness comment 3 regarding integration into clinical workflows and computation efficiency, as well as our response to weakness comment 2 regarding alignment.  We stress that virtual stain multiplexing does not require an alignment step at inference time and alignment was only necessary for construction of paired WSIs for model training.
> 4. Alternative analyses with different handling of the “unknown” class is presented in Appendix A. Figure 4 presents results with cells annotated by P1-P3 as “unknown” set to CD68+, instead of CD68- as in the main text. Figures 5-6 present results where cells annotated by the study pathologist are also set to CD68- (Figure 5) or CD68+ (Figure 6). We note that the trends of the results are robust to these choices, with the addition of VSM improving performance metrics in all cases.

---

> > ### Comment · Area_Chair_4Zsu · 2025-03-14
> >
> > Please update the final rating after appreciating the authors' responses.

---

### Official Review · Reviewer_Fv2Q · 2025-02-24

**Confidence:** 3
**Preliminary Rating:** 3
**Final Rating:** 3

**Summary:**

The paper presents an evaluation of the impact of virtually stained tissues to assists with detecting macrophages in tumor microenviroments of PD-L1 IHC pharmDx-stained samples of non-small cell lung cancer (NSCLC). It reports improved performance and inter-rater agreement on this task when raters have virtually stained samples (+ CD86, derived from PD-L1 stained samples) available.
The paper at hand builds on a previous study that introduced the methodological approach for virtual staining, with the focus in this paper being on the impact on the raters.

**Strengths:**

- the paper analyses a highly relevant aspect of pathology and especially virtual staining approaches - what impact they can actually make for a clinical application/(bio-)medical task; it motivates the clinical use-case well and proposes a plausible use case for virtual staining
- it carefully evaluates the decision for macrophages with "unknown" status, i.e., with unclear staining status (but see question below)
- the paper is well written and well structured, following a clear storyline and provides extensive visualizations.

**Weaknesses:**

The main methodological contribution has been presented in a previous study, leaving "just" the inter-rater assessment (which is still important!) Therefore, given this focus of the paper, certain aspects regarding the labeling and the evaluation are missing from the evaluation from my perspective:
- the inter-/intra-rater consistency is evaluated on the PD-L1 and on the VSM but -- to the best of my understanding -- not on the PD-L1 + CD68 stained data. Given that this labeling tasks seems to be inherently complex, it is not clear what the quality of the "ground truth reference" of the study pathologist is and how large the variability on the real, CD68-stained images can be.
- it is good to see that the authors used a rather long wash-out period to avoid that the pathologists remember specific cases; however, it is not discussed how much experience the raters have on this task (both for assessing macrophages on CD68 and on pure PD-L1 stained images.
- the annotation / labeling process for the study pathologist and the raters is described rather shortly, and many questions remain, including: Were specific instructions / training / examples provided? Which tool was used? Did the raters have a time limit or similar constraints?

**Detailed Comments:**

Additional/Minor comments:
- pg. 2: "tasks(Robert et al., 2023 [...]" - space missing
- Fig. 1: Potentially the space could be utilized better, with larger images for better visualization?
- Fig. 2.D: I struggled in understanding the message of this subfigure. Given that this is explained on Page 5, maybe it makes sense to place this figure later in the text?
- pg. 6: It could potentially improve clarity to highlight the different consensus types in the beginning of Sec. 3, i.e., highlight "majority consensus" vs. "full consensus". For this consensus task, were macrophages labeled as "unknown" counted as CD68- as described in the previous section?
- pg. 6: some of the evaluations / accumulations (e.g., Fig 3) did not become fully clear to me - I presume that the upper Row in Fig 3 describes the comparison with the reference from the study pathologist. Does "consensus" stand for majority vote? Potentially this information could be added in the figure caption.

**Justification Of The Final Rating:**

I would like to thank the authors for their detailed discussion in the rebuttal. In my initial review, I rated the paper as "borderline" as several aspects and choices were not clear, especially since the focus of the paper was on the validation, not on the introduction of a novel methodology. Many of these questions were answered in the rebuttal.

From my perspective, the paper and the investigation of the utility of virtual staining for specific applications is very interesting and could spark relevant discussions at MIDL. The general finding that it can improve cell differentiation for human readers and improve consistency may be relevant also in the context of semi-automatic ML pipelines.

That said, my rating of the paper remains at "borderline". I believe that an assessment of inter-rater variability on PD-L1 + real CD68 staining is still relevant, even or especially with the "unknown" class. Additionally, while pathologists may be familiar with PD-L1 staining, to the best of my understanding, TPS assessment does not necessarily involve an individual, per cell assessment, but distributions may also be estimated. It therefore seems imbalanced that guidelines/trainings were provided for the VSM staining assessment but not for the PD-L1-only staining.
Therefore, while I don't believe that these findings will change the direction of the study completely, these aspects weaken this validation-focussed paper.

For my rating, I did not factor in the use of (fully) automated detection pipelines for this task; however, for future research, I believe it would also be interesting to consider whether an automated detection algorithm benefits from VSM (due to additional learning signals or the integration of more data), which the authors may want to put into perspective with the performance of human raters using VSM.

**Justification Of The Preliminary Rating:**

The contribution of the paper at hand lies in the assessment of the improvement of inter-rater consistency and detection performance when using virtual staining; however, an important aspect of this is the quality of the ground truth that the "aided" annotation process is compared with. Following my questions above, I believe that the current evaluation may not be sufficient to get a full picture of the benefit of the proposed approach, and I would like to understand the reason for not including this aspect in the analysis as well.

**Questions To Address In The Rebuttal:**

While it is not surprising that the consistency increases as additional information is provided, the quality (actual detection performance) of the annotations is currently evaluated only on the annotations of a single rater. From my experience, this can severely limit the expressiveness of this assessment - can the authors comment on this limitation?

The biggest question for me for this paper whether any estimates / experiments have been conducted to understand the inter- and intra-rater consistency of the annotation of the "true" PD-L1 + CD68 stained tissue, as well as the labeling instructions, guidelines, tools, etc. and how they may impact the intra- / inter-annotator consistency. How could these effects potentially affect the study, and especially effect the ground truth?

Lastly, can the authors comment on why they force labels for the unknown class? Wouldn't it make sense to also consider this -- in some way -- as a separate category and evaluate it as such?

---

> ### Author Response · Authors · 2025-03-08
>
> We thank the reviewer for their comments and questions. Below, we provide our responses to the key points raised.
>
> **Weaknesses:**
>
> - We agree with the reviewer that the inter-rater assessment is important. We consider the rater to be an integral part of a virtual stain based assistive method, so we considered putting a significant effort into the evaluation important.
> - We acknowledge the challenge posed by high inter-observer variability in macrophage identification; however, this challenge arises primarily when using PD-L1 stain alone. Therefore, we have incorporated an additional CD68 stain, a reliable and specific marker of macrophages that does not stain any other type of cells. In addition, we incorporated “unknown” category for inconclusive cases to enhance the robustness and reliability of our ground truth. Furthermore, we have provided a detailed annotation workflow [Appendix B], including comparative image pairs that illustrate the distinction between CD68+ macrophages, CD68- cells, and ambiguous cases classified as "unknown."
> - We thank the reviewer for appreciating our diligence in ensuring a long wash-out period to avoid biases. The raters employed for the evaluation are all certified for PD-L1 reading which requires excluding immune cells such as macrophages from the calculation (PD-L1 TPS protocol - Agilent Technologies, 2021). We have added Appendix B (Data Annotation Workflow) to help clarify the training and guidance the evaluation raters received for performing their annotation tasks.
> - We agree adding more details on the annotation process can be helpful, to that end we have added Appendix B (Data Annotation Workflow) which provides these details. To the reviewer’s question specifically, the appendix includes the following information:
> For the PD-L1 only annotation phase pathologists “were instructed to annotate based on their prior knowledge and familiarity with PD-L1 staining”
> All annotations were performed using an in-house annotation software and “Upon enrollment, pathologists received training on using the in-house annotation software.”
> For phase 2 annotations using PD-L1 + VSM CD68 “The study pathologist provided an introduction to the characteristics of CD68 staining and instructed annotators on how to classify cells into CD68+, CD68-, or Unknown categories”
> Regarding time constraints we note that “All annotations tasks were done without time limits”
>
>
> **Detailed Comments:**
>
> - We thank the reviewer for detecting this typo, it is fixed in the new revision.
> - We have done our best to balance the amount of information we put into the visualizations with the space constraints. We note that Appendix E (Evaluation Regions) include larger visualizations of PD-L1, PD-L1 + CD68 IHC and PD-L1 + VSM CD68 of the regions used in the evaluation.
> - We thank the reviewer for the suggestion, in the new revision we have moved pushed figure 2 later in the text, as much as we were able.
> - We agree that highlighting the different types of consensus is important. Accordingly, we have to the end of the Methods section the definition “full-consensus labels were obtained for cells where the annotations of P1-P3 were the same”.
> In addition, we have added to the figure caption of Figure 3 explicit mention that the consensus presented in the figure is “majority-vote consensus”. We added similar clarification of consensus type to the captions of figure 2 and figure 8 (in the appendix). Cells classified by P1-P3 as “unknown” were set as CD68- in all performance metrics presented in the main text.
> - We have added explicit mention that the metrics were computed by comparison with the study pathologist’s ground truth annotations.
> In the methods section we write: “Performance metrics were computed for each of P1-P3, comparing their annotations to the ground truth annotations by the study pathologist.”. Similar language was also included to the expanded Performance Metrics appendix C, where we note.
> In addition, and as suggested by the reviewer, we added this information to the caption of Figure 3.
>
> **Questions To Address In The Rebuttal:**
>
> - We appreciate the reviewer’s concern regarding the ground truth. Please see our response under Weaknesses section second bullet.
> - We were guided in our analysis choices by the fact that clinicians performing the TPS scoring task must make a call regarding the cells included in the calculation. To account for the effect our choice on how to handle the unknown class on our results, we provided in the appendix several alternative analyses, demonstrating that the observed trends in improvement of pathologists’ performance when using VSM are preserved.

---

> > ### Comment · Reviewer_Fv2Q · 2025-03-12
> >
> > Thank you for providing the rebuttal and a revised version of the paper, which clarifies several questions and improves the quality and detail of the manuscript.
> >
> > Still, similarly to *reviewer Qfiq*, I am also missing an answer to why single reference by the study annotator is sufficient for the task at hand (e.g., a previous evaluation that describes the inter-rater/intra-rater agreement on PD-L1+CD68 stains. This strikes as particularly relevant given the "unknown" class, which indicates that there can be uncertainty and also a rather-specific decision threshold in the decision.
> > Can you motivate more clearly why a single rater is sufficient?
> > In our own work, we have seen again and again that "seemingly easy / clear-cut" tasks have a surprisingly high inter-rater variability, especially in digital pathology.
> >
> > A further concern relates to the annotation instructions: If the annotation for the original PD-L1 slides was only done based on "prior knowledge and familiarity of LD-L1 staining" vs. additional instructions for the VSM staining, would this not introduce a knowledge bias between the two settings? Can you comment on the experience of the pathologists with annotating macrophages on only-PD-L1 stained images?
> >
> > I agree with *reviewer Qfig* that, since the main contribution of this paper is the validation, the experimental setting and the selection of the reference standard is particularly essential. Furthermore, can you comment on whether the current performance of pathologists' rating using VSM is sufficient for clinical usecases (see also *reviewer Qfig*)?

---

> > > ### Author Response · Authors · 2025-03-13
> > >
> > > We appreciate the reviewer’s thoughtful feedback and the opportunity to further clarify our work. We hope our responses below will address the additional questions.
> > >
> > > - Our rationale for using a single reference pathologist is explained in our response to reviewer Qfiq, Question 6.
> > > Reproduced here for convenience:
> > > > This is an important point, and we appreciate the opportunity to clarify our reasoning for using a single ground-truth pathologist.
> > > > CD68 staining can sometimes generate patterns that are difficult to confidently associate with individual cells. To ensure a reliable ground truth, we considered two potential approaches: increasing the number of annotators or selecting a subset of cells that, given the additional macrophage-specific CD68 staining, could be classified with high certainty. We opted for the latter approach, prioritizing only those cells that could be unequivocally identified.
> > > > As demonstrated in the images in Appendix B, only cells that exhibited strong magenta CD68 staining in the vicinity of their nucleus were annotated as positive. Given these simplified classification rules, we believe a single expert pathologist was sufficient for establishing a robust ground truth.
> > > > Moreover, as discussed in the Introduction regarding the potential shortcomings of consensus scoring, we do not believe multiple raters would resolve the ambiguity in CD68 staining for “unknown” cells (Appendix B), consistent with the low agreement in unassisted macrophages detection.
> > >
> > >
> > >   - Regarding the decision threshold, we consider the “unknown” class to be a buffer, widening the decision boundary between positive and negative classifications. As described in Appendix B and shown in Figure 8A (Appendix D), the study pathologist used the “unknown” category liberally, electing to err on the side of certainty. Furthermore, alternative analyses in Figures 5–6 (Appendix A) show that assigning all “unknown” cases to either class does not alter the qualitative improvement trends reported in the main text. This consistency across different handling strategies suggests that the annotation choice is robust and reinforces that the observed improvements are driven by the virtual stain multiplexing method rather than specific annotation decisions.
> > >
> > >
> > >
> > > - We appreciate the reviewer’s concern regarding annotations instruction. While this point is briefly discussed in the manuscript, we acknowledge the need for further clarification.
> > >   - The evaluation pathologists involved in this study are certified and experienced in PD-L1 scoring, as it is part of their academic and clinical training. The PD-L1 IHC 22C3 pharmDx Interpretation Manual [PD-L1 TPS protocol - Agilent Technologies, 2021], which defines the FDA approved PD-L1 scoring process, explicitly instructs pathologists to exclude macrophages from scoring based solely on PD-L1 staining. This manual also provides examples to guide pathologists in distinguishing between tumor cells and immune cells. Additionally, the selected evaluation pathologists regularly perform PD-L1 scoring as part of their routine practice.
> > >   - In contrast, the VSM of PD-L1 and CD68 is not part of traditional staining, and viewing slides in a side-by-side manner (PD-L1 only together with VSM) is also not a standard practice. Therefore, a short training session was provided to ensure that pathologists were familiar with the multiplexed staining and could interpret it effectively.
> > >
> > >
> > >
> > > - Regarding your question about clinical use case, please see our response to reviewer Qfiq, Question 4, reproduced here for convenience:
> > > > We appreciate the question, as it addresses a crucial aspect of the clinical utility of virtual staining. It is important to reiterate that further model development and validation on larger datasets are required before this approach can be considered for clinical use. However, our findings already highlight a potential case where the TPS score could shift between zero and non-zero depending on the accurate exclusion of positively stained macrophages, as required by the PD-L1 interpretation manual. Such a shift could directly influence treatment decisions, reinforcing the significance of this method.
> > > > For instance, in ROI #6 (Appendix E), most PD-L1-positive cells were initially (using PD-L1 only slides) considered tumor cells, using our VSM, they can be easily identified as macrophages. From Figure 3A-B and Table 9 (Appendix C), we observe that both P1 and P2 would have potentially assigned a different clinical score with the additional information provided by the assistive layer. While not the primary focus of this study, existing literature suggests that mapping the immune system within the tumor microenvironment may provide valuable prognostic insights.

---

> > > > ### Comment · Reviewer_Fv2Q · 2025-03-14
> > > >
> > > > Thank you for the additional explanations. I have no further questions at this point in time.

---

### Official Review · Reviewer_PsQB · 2025-02-26

**Confidence:** 3
**Preliminary Rating:** 4
**Recommendation:** Poster

**Summary:**

The paper aims to identify macrophages in PD-L1 IHC stained NSCLC tissues.

Specifically, the authors used a virtual stain multiplexing approach that computationally adds CD68 macrophage markers to existing PD-L1 stained slides without physically restaining them via a U-Net  trained on 49 NSCLC whole slide images that had been sequentially stained with both PD-L1 and CD68 markers.

Evaluations show that inter-pathologist agreement can be improved with the virtual staining.

**Strengths:**

The paper has a strong medical significance, as accurate PD-L1 scoring requires proper exclusion of macrophages.

The improvement in inter-observer agreement is observable.

The use of sequentially stained CD68 IHC as ground truth provides an objective reference standard rather than relying solely on subjective pathologist opinions.

**Weaknesses:**

Major:
The study relies on a relatively small dataset of 49 NSCLC whole-slide images, and lacks external validation on datasets from different institutions, scanners, or staining protocols.  Moreover, the evaluation focuses primarily on agreement and detection metrics but offers limited analysis of false positives/negatives and their clinical implications.

Details regarding the alignment process between sequentially stained slides could have been more thoroughly explained.


Minor:
The paper employs a standard U-Net architecture without exploring alternative or more advanced architectures that might yield better performance for this specific application.

**Detailed Comments:**

Please consider including a brief discussion of the computational resources required for model inference.

Please consider adding a discussion to provide more specific guidance on how this approach might be integrated into existing digital pathology workflows and laboratory workflows.

**Justification Of The Preliminary Rating:**

The paper has a strong medical significance, the presentation is clear. However the evaluations is confined to a small scale dataset and  raise questions about generalizability. Adding a discussion to describe the limitation of evaluation can be helpful.

**Questions To Address In The Rebuttal:**

The paper has clear explained its method and experimental details.

**Special Issue:**

No

---

> ### Author Response · Authors · 2025-03-07
>
> We thank the reviewer for their comments and questions. Below, we provide our responses to the key points raised.
> **Weaknesses:**
>
> -	We updated the manuscript to expand on the dataset curation process leading to the curated dataset comprising 49 WSIs. The 49 selected cases are not a random dataset. Instead, this is a distillation of a 330 cases from which cases were selected to ensure that cases with TPS score covering the possible range are represented, as well as cases near the clinical threshold. While we recognize that validation on datasets from different labs, staining and scanners would be valuable, that was beyond the scope of our current research.
> We added a statement to the Conclusion section to that effect. Added to the Methods section: “The study pathologist reviewed 330 scanned NSCLC whole-slide images (WSIs) stained with the PD-L1 IHC 22C3 assay. From these, 49 cases were curated to cover the full TPS range and include cases with PD-L1 expression levels near clinically relevant thresholds.”
> The conclusion section now ends: “Further work would be needed to further develop and validate on a larger, more diverse dataset that includes sections stained and scanned by different labs.”
> -	In depth analysis of clinical implications of VSM FP/FN in practical use is beyond the scope of our current research. We have added a sentence to the introduction to indicate in which cases confusing macrophages and tumor cells could have more of a clinical effect: “In near threshold cases, misidentification of tumor cells and macrophages can lead to change in treatment decision.”
> -	We updated the methods section to include a description of the algorithmic pipeline used to align the WSI pairs: “Both WSIs in each pair were globally aligned with RANSAC-optimized Euclidean transform, using ORB features (Rublee et al., 2011), followed by DIS optical flow (Kroeger et al., 2016) to enable pixel-accurate training,”
> -	We chose the U-Net architecture as the backbone of our VSM architecture because of its record as a standard for medical image processing. Because of the labor intensive nature of manual evaluation process and the associated cost of testing any additional architecture we preferred a backbone we are confident in using. We note that while U-Net can be considered standard, our overall architecture including transformation to OD and separation of learning stain concentration and RGB color are not standard. We plan to explore alternative architectures in the future.
>
> **Detailed Comments:**
>
> -	We have added a brief comment on this subject to the discussion in the Conclusion section: “Inclusion of VSM in pathology workflows is computationally feasible. An NVIDIA RTX A5000 Laptop GPU can virtually stain an X40 field of view (250um radius) in under a second, enabling real-time use, while an NVIDIA Tesla V100 GPU achieves the same in 0.5 seconds, supporting efficient parallelized offline processing of WSIs.”
> -	How best to employ virtual staining methods in general, and VSM specifically, in practice is a wider research question beyond the scope of the current evaluation. We hope to explore this question in future research.

---

> ### Comment · Area_Chair_4Zsu · 2025-03-14
>
> Please update the final rating after appreciating the authors' responses.

---

### Official Review · Reviewer_Qfiq · 2025-02-28

**Confidence:** 4
**Preliminary Rating:** 2
**Final Rating:** 3

**Summary:**

The authors validate a previously published model, which can generate a PD-L1+CD68 stained whole slide image, from a whole slide image stained only with PD-L1. The model is validated in an observer study with 3 observers, using kappa and detection metrics for quantitative comparison. The observers first detected and classified cells on the PD-L1 WSIs, and after a wash-out period of 6 months on the PD-L1 + AI generated CD68 WSIs. Detection performance is calculated with respect to a ground truth set by a single pathologist on actual PD-L1+CD68 stained images.

While the general premise of the method is interesting, the validation fails to impress due to a low-quality ground truth in a high inter-observer variability task.

**Strengths:**

- The observer study is generally well designed, with an adequate number of WSIs included and a long wash-out period to avoid bias.
- The premise of virtually staining images is interesting, and the paper demonstrates that to some extent this allows pathologists to generate more insight from a PD-L1 stained image.
- The study also uses a variety of metrics to quantify both agreement and performance, which allows the reader to judge the results well.

**Weaknesses:**

The main weakness of the paper, especially considering it is a validation paper, is the ground truth. Although the ground truth is based on actual CD68 staining, the cell locations and CD68+/- labels are set by a single pathologist only. This goes directly against the arguments of the authors in the introduction (and the main motivator for the paper method), where they state that inter-observer agreement is low. It would have been much stronger if the ground truth was set by a consensus panel of at least three pathologists.

For instance, in Figure 7/Appendix C, if you pool annotators BCD, there are more annotations than the ground truth annotator, meaning more cells are detected. Are those extra cells missed by the ground truth annotator?

In this light, it is also crucial to clearly present the AI standalone performance. Does the AI over/under predict cells? This directly influences reader performance.

It would also be instructive to see how well an AI model trained to detect cells directly from PD-L1, using the PD-L1+CD68 ground truth, performs with respect to the pathologists. Right now the performance of the AI staining accuracy and the pathologist reading performance are mixed. If I understand correctly, pathologists are not trained to detect CD68+/- from PD-L1 slides, so it is not surprising that their performance and agreement is low in that case.

Finally, quite a large part of the paper is dedicated to improvements in agreement, but in general when you introduce AI assistance, agreement will improve, so that conclusion is not very new or surprising.

The paper could become stronger by improving the reference standard, and spending more words/analysis on detection performance instead of agreement.

**Detailed Comments:**

It is not completely clear exactly how all metrics are calculated. It could be nice to clearly explain how you set true/false positive/negative, and how your metrics follow from them.

Figure 2 is slightly confusing, since it mixes the ground truth annotation flow, with the flow that the observers see. I think it might be clearer to separate those out in the Figure. It is clearly explained in the text though.

As I alluded to before, the conclusion is quite long for not that many conclusions to be made. I think the paper could benefit from more detailed and precise explanation of methodology (and perhaps on the AI system/performance) instead of some conclusion paragraphs.

**Justification Of The Final Rating:**

Initially I rated the paper a 'weak reject', mainly because of a suboptimal ground truth annotated by a single reader on a task which seems not clear cut. Especially considering that the main contribution of the paper is validation of an existing technique, this felt out of place. It was also difficult to understand these subtleties clearly from the paper.

After the reviewer discussion period, where the authors went into detail regarding these limitations, I have decided to upgrade my rating to 'borderline'. The main conclusion of the study - that virtual staining improves inter-observer variability and classification performance - will likely hold in a more thoroughly designed setting, although the precise numbers need to be taken with a grain of salt w.r.t. clinical practice.

However, after the discussion period, I still feel as if the study would have only gained from using multiple ground truth observers. In this light, I still do not see a clear reason why this was not done. I would highly recommend the authors to add this in their future work. This strengthens both the ground truth w.r.t. which the performance metrics are calculated, and it puts the inter-observer variability metrics in context. With the last point I mean that the authors could show what the inter-observer variability difference is between PDL1+real CD68 staining vs. PDL1+virtual CD68 staining.

**Justification Of The Preliminary Rating:**

While the premise is interesting, this paper does not introduce the AI model which was previously published. The main contribution, therefore, is the validation. Given the weak ground truth, it is hard to make definitive conclusions on the added value in clinical practice.

**Questions To Address In The Rebuttal:**

Can you explain the choice for using only 1 pathologist as reference standard?

---

> ### Author Response · Authors · 2025-03-07
>
> We thank the reviewer for their comments and questions. Below, we provide our responses to the key points raised.
>
> **Weaknesses:**
> * We acknowledge the challenge posed by high inter-observer variability in macrophage identification; however, this challenge arises primarily when using PD-L1 stain alone. Therefore, we have incorporated an additional CD68 stain, a reliable and specific marker of macrophages that does not stain any other type of cells. In addition, we incorporated “unknown” category for inconclusive cases to enhance the robustness and reliability of our ground truth. Furthermore, we have provided a detailed annotation workflow [Appendix B], including comparative image pairs that illustrate the distinction between CD68+ macrophages, CD68- cells, and ambiguous cases classified as "unknown."
> * The evaluation pathologists classified **only the pre-defined cells** that were originally annotated by the study pathologist. This ensured that all annotators assessed the exact same set of cells across all cases, maintaining consistency in the total number of annotated cells for each pathologist.
> * If the reviewer’s comment pertains specifically to the increased number of positive annotations made by the evaluation pathologists, this does not indicate missed detections by the study pathologist. Instead, these additional annotations reflect false-positive classifications by the evaluation pathologists, as evidenced by their lower classification performance.
> * We note that the AI model presented in this study does not predict individual cells but rather generates a virtual stain multiplexing (VSM) layer as an assistive tool. The study does use an automated cell detection algorithm as an assistive tool to facilitate cell detection. Eventually, all cells included in the evaluation were validated by the study pathologist. We’ve added clarification in Appendix B
> * Our current model does not perform automatic cell classification but rather provides an assistive VSM layer that significantly enhances pathologists' ability to classify macrophages. Future work may include macrophage classification models, for which this VSM approach could serve as an interpretable confidence-building layer.
> * The reviewer is correct in noting that our findings reflect a combination of model performance and pathologist reading performance. We now clarify this distinction more explicitly in the manuscript and suggested that full-consensus analysis may serve as a reasonable proxy for assessing model performance. Furthermore, we highlight an anecdotal false-positive case in Figure 2C, where a full-consensus annotation of a cell as CD68- is attributed to model performance, as actual magenta staining is visible in the sequential IHC staining (Figure 2B).
> * Regarding pathologist training - as stated in the PD-L1 IHC 22C3 pharmDx Interpretation Manual (Agilent Technologies, 2021), pathologists are instructed to “Score only viable tumor cells. Exclude all other cells from scoring: infiltrating immune cells...” This emphasizes the relevance of our work, as our study shows that, without assistance, pathologists exhibit poor performance on this task. These findings further reinforce the need for tools like VSM to improve accuracy and consistency in macrophage detection.
> * As noted by the reviewer it is true that AI assistance generally improves agreement. Our study goes beyond demonstrating this expected outcome. We specifically focus on a task—macrophage detection in PD-L1-stained slides—where inter-observer variability is exceptionally high. Traditional consensus annotation is unreliable and can't serve as ground truth for certain extremely noisy tasks. By using VSM, we not only improve agreement but also demonstrate a substantial enhancement in macrophages detection accuracy. Our findings highlight that VSM meaningfully reduces false classifications and enhances pathologist performance, reinforcing its potential value in clinical practice.
>
> **Detailed comments:**
> * Per reviewer suggestion, we have provided a detailed explanation of the performance metrics, including precision, recall, specificity, and F1-score, in Appendix C [p.8]. In addition we have added explicit mention that the metrics were computed by comparison with the study pathologist’s ground truth annotations.
> * We believe it is important to thoroughly discuss the implications of our findings. Specifically, our study highlights the limitations of traditional consensus annotation in this context and demonstrates how VSM improves pathologist performance, which could critical insights for both research and clinical applications. The ultimate goal of our model is to assist pathologists in making more reliable assessments, reducing variability, and improving diagnostic precision.
> * Methodological details, the technical aspects of the AI model development have been extensively described in our previous publication (Ben-David et al., 2024).

---

> > ### Comment · Reviewer_Qfiq · 2025-03-11
> >
> > Thanks for the detailed reply to all the comments, and for the textual edits. Especially Appendix B makes the whole study flow much clearer!
> >
> > I see also from your comments that I misunderstood some specifics, such as the fact that all inter-observer pathologists only classify predetermined cells. That indeed reduces some of my concerns, but perhaps also makes your exact clinical goal a bit more confusing to me. I have a few extra questions for the discussion period in which we can hopefully clarify that.
> >
> > 1. The fact that you use only a single ground-truth pathologist suggests to me that if you have access to both PD-L1 and CD68 at the same time, assigning the classes is an easy task, with low inter-observer variability. Is that correct?
> > 2. If yes, then am I right in assuming that, since all inter-observer pathologists see the same AI-stained CD68, all readers should have very low inter-observer variability (i.e. at least 0.8 kappa or higher)? What surprises me, for instance, is that in fig 8 pathologist D scores almost 1000 cells positive, whereas B and C score only around 100 cells positive.
> > 3. If no, then do you agree that adding extra ground-truth pathologists is essential?
> > 4. If I assume for now that the answer to 1. is yes, then the AI performance determines for a large part the clinical performance. For the reader to determine whether this is clinically useful, could you therefore indicate whether the performance (w.r.t. to the ground-truth pathologist) is sufficiently high for clinical practice? Or in other words, is the virtual staining good enough to forego actual staining?
> > 5. Could you explain why you chose to preset cell locations? Is there not additional variability in clinical practice stemming from nuclei detection? (which depends on the accuracy of the AI staining)
> > 6. In any case, since you did not clearly come back to that question: could you clearly motivate the choice for a single ground-truth pathologist?

---

> > ### Author Response · Authors · 2025-03-13
> >
> > We sincerely appreciate the reviewer’s thoughtful feedback and engagement with our work. Your comments have helped us refine the manuscript, improve clarity, and better articulate key aspects of our study. We hope that our responses below will further clarify any remaining questions.
> >
> >
> > **Reviewer questions 1&2:**
> >
> > We apologize for not explicitly mentioning in the figure caption that Figure 8B–D present the distributions of the evaluation pathologists’ annotations performed without any assistive layer, meaning they relied **solely on PD-L1 staining**. The goal was to highlight the high variability in annotations under these conditions, exemplifying why consensus annotation cannot reliably serve as ground truth in this context. With the addition of virtual stain multiplexed (VSM) CD68 the kappa score improved dramatically, as seen in Table 1. we thank the reviewer for the opportunity to clarify this editing mistake and will correct it in the camera ready manuscript.
> >
> > Regarding your first question: The answer is Yes, when both PD-L1 and CD68 staining are available, macrophage classification becomes a mostly straightforward task (as illustrated in Appendix B). For cases where annotations were not straightforward, we introduced the unknown class in the ground truth, ensuring a reliable reference standard for the evaluation.
> >
> >
> > **Reviewer question 4:**
> >
> > We appreciate the reviewer’s question, as it addresses a crucial aspect of the clinical utility of virtual staining.
> > It is important to reiterate that further model development and validation on larger datasets are required before this approach can be considered for clinical use. However, our findings already highlight a potential case where the TPS score could shift between zero and non-zero depending on the accurate exclusion of positively stained macrophages, as required by the PD-L1 interpretation manual. Such a shift could directly influence treatment decisions, reinforcing the significance of this method.
> >
> > For instance, in ROI #6 (Appendix E), most PD-L1-positive cells were initially considered tumor cells but can be easily identified as macrophages using our VSM. From Figure 3A-B and Table 9 (Appendix C), we observe that both P1 and P2 would have potentially assigned a different clinical score with the additional information provided by the assistive layer.
> > While not the primary focus of this study, existing literature suggests that mapping the immune system within the tumor microenvironment may provide valuable prognostic insights.
> >
> >
> > **Reviewer question 5:**
> >
> > Biological variability arises from multiple factors, and while nuclei detection can introduce some level of variation, it is a relatively well-solved problem in computational pathology.
> >
> > In this study, our primary focus was on the specific challenge of macrophage detection. By pre-setting cell locations, we aimed to cancel out variability stemming from nuclei detection, allowing us to isolate and evaluate the impact of virtual staining on macrophage classification.
> >
> > Additionally, it is important to note that the AI model does not classify cells but rather predicts CD68 staining on the pixel level, there is no explicit notion of cell locations during training. Cell annotations are used only for evaluation and do not influence the training process. Therefore, variability in nuclei detection would not affect model performance during training.
> >
> >
> > **Reviewer question 6:**
> >
> > This is an important point, and we appreciate the opportunity to clarify our reasoning for using a single ground-truth pathologist.
> > CD68 staining can sometimes generate patterns that are difficult to confidently associate with individual cells. To ensure a reliable ground truth, we considered two potential approaches: increasing the number of annotators or selecting a subset of cells that, given the additional macrophage-specific CD68 staining, could be classified with high certainty. We opted for the latter approach, prioritizing only those cells that could be unequivocally identified.
> >
> > As demonstrated in the images in Appendix B, only cells that exhibited strong magenta CD68 staining in the vicinity of their nucleus were annotated as positive. Given these simplified classification rules, we believe a single expert pathologist was sufficient for establishing a robust ground truth.
> >
> > Moreover, as discussed in the Introduction regarding the potential shortcomings of consensus scoring, we do not believe multiple raters would resolve the ambiguity in CD68 staining for “unknown” cells (Appendix B), consistent with the low agreement in unassisted macrophages detection.
> >
> > In addition, we added alternative analyses [Appendix A] to validate the robustness of the improvement trends by assessing different approaches to handling “unknown” annotations.

---

> > > ### Comment · Reviewer_Qfiq · 2025-03-13
> > >
> > > Thanks for the additional detailed replies, from my side this clarifies my remaining questions.

---

### Author Rebuttal · Authors · 2025-03-07

**Rebuttal:**

We sincerely appreciate the reviewers’ thoughtful comments and suggestions, which have greatly contributed to improving our manuscript. In revising the paper, we carefully considered all feedback, addressing ambiguities and incorporating suggested additions wherever possible.

We have added more detailed information and additional technical details on the construction of our dataset, as well as some discussion on the limitations of our evaluation and its potential for practical use. Additionally, we expanded the appendices to provide a detailed description of the cell annotation process and formal definitions of the evaluation metrics. While we strived to accommodate all requests, some fell beyond the scope of our current research.

Detailed responses to each reviewer’s comments are provided separately.

We are grateful for the reviewers' insightful input and the opportunity to refine our work, and we thank them for their time and effort in reviewing our submission.

**Supporting Material:**

/attachment/1b31732b86b70ccbd13b93b706e159f909585dc7.pdf

---

### Meta-Review · Area_Chair_4Zsu · 2025-03-20

**Recommendation:** Accept (Poster)
**Confidence:** 4

**Metareview:**

The manuscript initially received 4 reviews (weak reject, weak accept, borderline, borderline) and enjoyed quite thorough post review/rebuttal discussion, resulting in an updated rating of (borderline, weak accept, borderline, weak accept).
This positive trend in the review scores suggests that, indeed, several shortcomings of the initial manuscript were addressed by responses; primarily this pertains to concerns around single reader ground truth and to some extent the small dataset that was being used in this study. Other concerns, such as short comings of the study design (such as the estimation of inter-rater variability on real staining) which some reviewers felt would have been best addressed by multiple reader ground truth and others feeling it would be more appropriately handled by the study participants.

Overall, after a quite thorough read of all information available to me (including the manuscript, reviews, responses, and discussion) my assessment is that - despite some favorable reviews - this manuscript is indeed perfectly borderline and I can see myself going either way. Let me elaborate: While I certainly appreciate the importance and wealth of human-AI interaction studies, of which this is one while not labeled as such, the value and importance of these studies derives from the study design and its execution. In this case, the manuscript focuses exclusively on this user-centered evaluation without making algorithmic contributions (use of a previously published method) which puts additional emphasis on the necessity of designing a reasonable study. While some aspects of the study design are well thought out (such as the wash-out period of 6 months to reduce memory effects, which I will say adds complexity to the study design and effort) there are also clear weaknesses. For example: the three pathologists used for the evaluation will most likely have been from the authors' direct network (because retention rates in studies are low, especially in uncompensated and long wash out period studies); this is to the detriment of the conclusions because the participants will likely be aware of the scientific developments and thus not necessarily impartial. Further, the low sample size prevents any meaningful statsitical analysis of potential failure modes of the algorithm and its implications on user behavior, and the analysis presented here also fails to more carefully  highlight those challenges and the exact human-AI scenarios that require more deliberate study. There is more that could be said about the shortcomings of the study design.
In conclusion, given the review outcome that marginally skews favorably, and given that I strongly feel that human-AI interaction work is underrepresented in current MICCAI/MIDL contexts, I will recommend acceptance but strongly urge the authors to take more seriously study design as in any more HCI focused venue, this would have been an immediate rejection.